

# Properties of individual contrails: A compilation of observations and some comparisons

Ulrich Schumann[1], Robert Baumann[1], Darrel Baumgardner[2], Sarah T. Bedka[3], David P. Duda[3], Volker Freudenthaler[4], Jean-Francois Gayet[5], Andrew J. Heymsfield[6], Patrick Minnis[7], Markus Quante[8],
Ehrhard Raschke[9], Hans Schlager[1], Margarita Vázquez-Navarro[1], Christiane Voigt[1,10], Zhien Wang[11]

[1] Deutsches Zentrum für Luft- und Raumfahrt, Institut für Physik der Atmosphäre, Oberpfaffenhofen, 80334, Germany
[2] Droplet Measurement Technologies Inc, Boulder, CO USA
[3] Science Systems and Applications, Inc, Hampton, Virginia, USA
[4] Ludwig-Maximilians-Universität, Meteorologisches Institut, Munich, Germany
[5] Laboratoire de Météorologie Physique, CNRS, Clermont-Ferrand, France
[6] National Center for Atmospheric Research, Boulder, Colorado, USA
[7] NASA Langley Research Center, Hampton, Virginia, USA
[8] Helmholtz-Zentrum Geesthacht, Institute of Coastal Research, Geesthacht, Germany
[9] Max Planck Institute for Meteorology and University of Hamburg, Hamburg, Germany
[10] Johannes Gutenberg-Universität, Institut für Physik der Atmosphäre, Mainz, Germany
[11] University of Wyoming, Department of Atmospheric Sciences, Laramie, Wyoming, USA

*Correspondence to*: Ulrich Schumann (ulrich.schumann@dlr.de)

**Abstract.** Mean properties of individual contrails are characterized for a wide range of jet aircraft as a function of age during their lifecycle from seconds to 11.5 hours (7.4 to 18.7 km altitude, -88°C to -31°C ambient temperature), based on a
compilation of about 230 previous in-situ and remote sensing measurements. The airborne, satellite, and ground-based observations encompass exhaust contrails from jet aircraft since 1972, and a few older data for propeller aircraft. The contrails are characterized by mean ice particle sizes and concentrations, extinction, ice water content, optical depth, geometrical depth, and contrail width. Integral contrail properties include the cross-section area and total number of ice particles, total ice water content, and total extinction (area-integral of extinction) per contrail length. When known, the
contrail-causing aircraft and ambient conditions are characterized. The individual datasets are briefly described, including a few new analyses performed for this study, and compiled together to form a "contrail library" (COLI). The data are compared with results of the Contrail Cirrus Prediction model CoCiP. The observations confirm that the number of ice particles in contrails is controlled by the engine exhaust and the formation process in the jet phase, with some particle losses in the wake vortex phase, followed later by weak decreases with time. Contrail cross-sections grow more quickly than
expected from exhaust dilution. The cross-section integrated extinction follows an algebraic approximation. The ratio of volume to effective mean radius decreases with time. The ice water content increases with increasing temperature, similar to non-contrail cirrus, while the equivalent relative humidity over ice saturation of the contrail ice mass increases at lower temperatures in the data. Several contrails were observed in warm air above the Schmidt-Appleman threshold temperature. The "emission index" of ice particles, i.e. the number of ice particles formed in the young contrail per burnt fuel mass, is
estimated from the measured concentrations for estimated dilution; maximum values exceed $10^{15}$ kg$^{-1}$. The dependence of





the data on the observation methods is discussed. We find no obvious indication for significant contributions from spurious particles resulting from shattering of ice crystals on the microphysical probes.

## 1 Introduction

Contrails are well known as line-shaped ice clouds behind aircraft containing many small ice particles (Schumann, 2005).
Contrails form mainly from water vapor and particles emitted by the aircraft engines in cold air (Schumann and Heymsfield, 2016). After formation, contrails grow in width and ice water content (IWC) in ice-supersaturated air (Brewer, 1946). With aging, the initially linearly shaped contrails deform and merge with other contrails and cirrus to form "contrail cirrus" (Liou et al., 1990; Sassen, 1997; Schumann, 2002). Individual contrails may be observable for hours (Minnis et al., 1998; Vázquez-Navarro et al., 2015). Contrails and contrail cirrus have climate effects, which are relevant compared to other climate effects of aviation (Penner et al., 1999; Lee et al., 2009; Boucher et al., 2013; Brasseur et al., 2016).

Much has been learned from measurements about contrail properties and their climate effects (Heymsfield et al., 2010). Various models have been developed to compute contrail properties for individual contrails (Paoli and Shariff, 2016) and for contrail cirrus (Burkhardt and Kärcher, 2011). To predict and diagnose contrail formation and lifecycle, Schumann (2012) developed the Contrail Cirrus Prediction model (CoCiP), which uses various approximations to simulate a large ensemble of contrails efficiently.

Still, many questions are open: How realistic are the contrail model results? Do the models simulate the lifecycle with local and "integral" (cross-section integrated) properties (Jeßberger et al., 2013; Lewellen, 2014) similarly as observed? Can one verify large-eddy simulation predictions which show that only a small fraction of contrail ice particles survives the wake vortex regime (Unterstrasser, 2016) with measurements? Can one exclude that additional ice particles form later in contrails, e.g., for high humidity and strong upward motion (Jensen et al., 1998a; Spinhirne et al., 1998)? Which ice particle habits are formed in contrails and how do they affect radiative climate forcing (Liou, 1998; Markowicz and Witek, 2011)? How important are fallstreaks from contrails and what limits the resultant contrail depth (Knollenberg, 1972; Unterstrasser and Gierens, 2010b)? How do contrails influence subsequent cirrus formation, i.e., will the combustion particles emitted reactivate as cirrus crystals after they have been conditioned as contrail crystals (Zhou and Penner, 2014; Schumann and Heymsfield, 2016)?

For an improved understanding of contrail formation and evolution and for model testing and improvement, contrail observational data are needed. Contrail observations have been reported since 1915 (Ettenreich, 1919). Early measurements behind a propeller driven aircraft (see Appendix A) provided evidence that contrail ice formation requires liquid saturation to start (aufm Kampe, 1943), consistent with the now established Schmidt-Appleman criterion (Schumann, 1996). Those observations, with ice particles collected on impactors and halo observations, also provided the first information on the size and shape of contrail and cirrus ice particles (Weickmann, 1945). Measurements of contrail ice with optical particle counters





were introduced by Knollenberg (1972). Besides "exhaust contrails" forming from engine emissions, "aerodynamic contrails" forming because of adiabatic cooling near curved surfaces of the aircraft (Gierens et al., 2009; Kärcher et al., 2009b; Gierens et al., 2011; Jansen and Heymsfield, 2015) have also been identified. In addition to the question of when contrails form, one needs to understand the entire lifecycle of contrails, from initiation until final sublimation or precipitation
(Schumann and Heymsfield, 2016).

The critical testing of models requires reliable quantitative data, not only on the contrail and plume properties, but also on the contrail age and the generating aircraft, on the atmosphere conditions in which the contrail formed, and on the observational methods. This paper considers individual contrails with known ages, because these should provide more specific model tests.

For this study, we have collected data on individual contrails from various measurements reported in previous publications, partly with additional information collected from the experimenters and some from re-analysis of the data, and compare them to CoCiP results. The results extend a previously constructed figure with data in Schumann and Heymsfield (2016), and include data from recent measurements during ML-CIRRUS (Voigt et al., 2016). In addition to in-situ data from contrail measurements, remote-sensing data from ground-based or airborne lidar and spectroradiometer, satellites, cameras,
and visual observations are also included.

The results are compiled in a "contrail library" (COLI), i.e., a data set containing mean values with some error bounds representing observed properties, and information on the measurement methods, which may be useful for contrail research beyond the present study. Most in-situ data include ice particle size-spectra measured with optical particle spectrometers (Baumgardner et al., 2011; Wendisch and Brenguier, 2013), from which we take mean values such as ice particle number
concentration and IWC. Remote sensing data provide mean properties like width and optical depth $\tau$ (Spinhirne et al., 1998; Duda et al., 2004). Future studies should include exhaust and aerosol properties, data of aerodynamic contrails, comparisons of the data with other model results, and contrail cirrus.

## 2 Contrail-library parameters

The contrail library, COLI, is mainly an Excel table (see supplement) that contains a list of contrail observations. For each
observation, we collected available information on the aircraft and engines used, fuel (here kerosene in all jet cases), atmosphere, instrumentation, projects and references, and mean contrail properties. Tables 1 to 6 list and define the corresponding parameters. The list of parameters should be extended when more data (e.g., for alternative fuels) become available. Because of the variety of sources, the information available for individual observations covers subsets of the set of all parameters. Here we collect data from the investigations listed in Table 7, sorted by year of observation. We also include
older data, discussing instrument aspects, because the number of observations with sufficient information is small and we try to cover a wide range of ages and atmospheric conditions.





Mean values are plotted versus age in Figure 1 and Figure 2. For reference to the theoretically expected order of magnitude and for consistency checks, the plots also depict mean values of results from a CoCiP-CAM study as described in Schumann et al. (2015). CoCiP is a Lagrangian model simulating the lifecycle of individual contrail segments for a given contrail-generating aircraft and ambient meteorology, in this case as provided by the climate model CAM; see Appendix B.

CoCiP results have previously been compared to observations (Voigt et al., 2010; Schumann, 2012; Jeßberger et al., 2013; Schumann et al., 2013a; Schumann et al., 2013b; Voigt et al., 2016), including remote sensing observations for a large set of individual contrails (with unknown contrail ages) from Iwabuchi et al. (2012) and contrail-cirrus observations (Minnis et al., 2013; Schumann and Graf, 2013).

Original measurement data or links to the data banks containing the data were collected when available. The COLI table

includes information, when known, on how to access at least part of the original measurement data. Data archives are available for the NASA aircraft field experiments (https://espoarchive.nasa.gov/), the UK BAe146 aircraft (http://badc.nerc.ac.uk/), and for the Deutsches Zentrum für Luft- und Raumfahrt (DLR) research aircraft, Falcon and HALO (https://halo-db.pa.op.dlr.de/). Meteorological data are partly derived from nearby radiosondes (e.g., from http://weather.uwyo.edu/upperair/sounding.html) and from numerical weather prediction (NWP) analyses, e.g., from the

ERA interim reanalysis data of the European Center for Medium-Range Weather Forecasts, ECMWF (Dee et al., 2011).

Air traffic data are usually not available publically. Some projects collected flight data from air traffic control or ground-based observations (Schumann et al., 2013a; Schumann et al., 2016). Projects with air traffic subsets (since 2005) are identified in the table when available. Some traffic data for the USA and southern Canada were collected for 2000-2005 by Garber et al. (2005) (http://www-pm.larc.nasa.gov/prod/flttrkdbase/). For Germany, traffic data are collected and archived by

the Deutsche Flugsicherung (DFS) since the year 2006, for the more recent projects. The Aviation and Climate Change Research Initiative ACCRI (Brasseur et al., 2016), provided a global waypoint dataset for the year 2006 to scientists involved in the ACCRI project.

Aircraft properties (size, mass, speed, fuel consumption, propulsion efficiency) were obtained from various sources including the Base of Aircraft Data (BADA) (EUROCONTROL, 2009). Some engine properties, such as fuel consumption

and emissions at surface pressure, can be obtained from the International Civil Aviation Organization (ICAO) Aircraft Engine Emissions Databank (http://www.easa.europa.eu/document-library/icao-aircraft-engine-emissions-databank). Because of incomplete information (e.g., on aircraft weight), some data had to be estimated. Comments in the table explain these estimates. Mistakes when reading data from various sources cannot be fully excluded and users should check and verify original sources when details matter. Additional information may lead to revised library variants in the future.

For analysis of contrail data, an estimate of dilution of exhaust gases in the wake is often needed. A dilution factor $N_{dil}$ can be defined, which is the ratio of plume air mass to fuel mass burnt per flight distance. Here, we use

$$N_{dil} = 7000 \, (t/t_0)^{0.8}, \tag{1}$$





with $t_0 = 1$ s, for plume ages t of about 1 to 10000 s (Schumann et al., 1998). This relationship has been derived empirically from a set of airborne exhaust measurements behind various jet aircraft in cruise conditions. The relationship approximates exhaust dilution observations within a factor of about 3. It may underestimate dilution in contrails. Mixing and dilution depend on turbulence in the aircraft wake and in the ambient air, and is dependent, among others, on aircraft parameters,

shear and stratification (Paoli and Shariff, 2016). For this reason, related parameters are included in COLI.

Also, we often refer to the ratio

$$C = r_{vol}/r_{eff} = (r_{area}/r_{vol})^2 \qquad\qquad\qquad (2)$$

between effective, volume, and area mean radii, which is a consequence of the radius definitions, see Table 6 (Schumann et al., 2011; Schumann and Heymsfield, 2016). The value of C is one for spherical and monodispersed particles; it can be larger

than one for non-spherical ice particles; and can be far smaller than one for clouds with a wide particle size distribution (PSD) of number density which decreases with particle size d as $d^{-n}$, where n is between 0 and 4. The corresponding volume size distribution varies as $d^{3-n}$ for spherical particles, and must decrease at least with power -4 at high d to have a finite integral. For such PSDs, the volume size distribution reaches its maximum value for large sizes than the number size distribution. C becomes small when this size ratio becomes large. The parameter C is important because the projected area

density of ice particles in a cloud is $A_{ice} = C (9\,\pi/16)^{1/3}\, n_{ice}^{1/3}\, V_{ice}^{2/3}$ for given number and volume densities $n_{ice}$ and $V_{ice}$ (see Eq. 11 in Schumann et al. (2011)), and $A_{ice}$ becomes small for small C. Models deriving optical cirrus properties for given $n_{ice}$ and $r_{vol}$ (from IWC and $n_{ice}$) have to parameterize C to estimate $r_{eff} = r_{vol}/C$ (Lohmann, 2002; Liu et al., 2007; Penner et al., 2009; Schumann, 2012).

Below, the various observations are identified by the project name, if available. New project names were invented in

case of missing project names or for results from several projects. Moreover, we distinguish between in-situ and remote sensing observations.

The following section explains the data selection from the observations, first for in-situ data and then for remote sensing data (with some overlap). Some related projects are discussed together; the last column in Table 7 provides the link between the projects and the subsections. Each subsection identifies the sources of individual data, the conditions under which the

contrails formed, and how they were observed. This includes, e.g., information on whether the contrails formed in clear air or in cirrus, in ice supersaturated or subsaturated conditions. The intent of this description is to give the reader sufficient context and references for assessing and selecting suitable data for analyses.





## 3 Contrail-library description

### 3.1 In-situ observations of contrail properties

#### 3.1.1 PMS: First particle-size optical spectrometry in contrails

The first set of contrail ice particle data from in-situ observations became available from Knollenberg (1972). Knollenberg is

known as a pioneer of optical particle size spectrometers and founder of the company Particle Measurement Systems, PMS. The reader is referred to Wendisch and Brenguier (2013) for a tribute to Dr. Robert Knollenberg. The measurements were performed using a Learjet observing the contrail of the National Center for Atmospheric Research (NCAR) Sabreliner. The contrail formed in relatively warm and highly ice-supersaturated air. A photo shows that the contrail formed cirrus uncinus (Heymsfield, 1975) with large ice particles sedimenting in fallstreaks more than 500 m below the contrail core. The particle

spectrometer, a one-dimensional optical array probe (OAP) (Knollenberg, 1970), counted and sized ice particles of 75 to 1200 μm diameter at two flight levels embracing the contrail core. Hence, the many small ice particles below 75 μm diameter were not measured. The study quantifies the ice particle concentration $n_{ice}$ of the larger crystals, their ice water content IWC, and provides estimates of the contrail width W and depth D, and the total ice mass TWC and total number of ice particles $N_{ice}$ per unit contrail length, at ages of 1080 s and 2340 s, for a contrail associated with uplift from mountain-

induced waves. At 1080-s age, the contrail has removed about 41 kg m⁻¹ of ice mass from the ambient, ice-supersaturated air, more than 26000 times the mass of water emitted from the engines (1.6 g m⁻¹). The total number of (large) ice particles reached $1.5 \times 10^{10}$ m⁻¹. This value is far below other values, as we will see. Both values (and also W) decreased between 1080 s and 2340 s, without indications of fresh ice particle nucleation in the contrail. IWC, W, and D data are shown in Figure 1. The $n_{ice}$ and particle size values are not included in the plots because of missing small ice particles.

We note that the reported ambient temperature of -38.2°C exceeds the Schmidt-Appleman criterion (SAC) threshold temperature of -40.1°C for contrail formation by at least 1.9 K, even if one uses ambient pressure p, for the given flight level (29500 feet; i.e., p = 307.8 hPa instead of the reported 326 hPa), assumes ambient humidity at liquid saturation, a high water emission index $EI_{H2O}$ = 1.25, a low combustion heat $Q_C$ = 43 MJ kg⁻¹, and a high propulsion efficiency η = 0.35. The discrepancy may be explained by temperature measurement errors (Knollenberg, 1972) (e.g., measurements at altitudes

slightly below the contrail formation level), or by added plume humidity from ambient cirrus ice particles entrained into the exhaust plume. Also, the engine may have been still cooler during non-steady operation than at steady-state, absorbing some of the combustion heat. Further possibilities are discussed in Section 4.4.

#### 3.1.2 CSAE: High particle concentrations in young contrails

In a short note, Baumgardner and Cooper (1994) presented airborne measurements of ice particle size spectra in a young

contrail of a Learjet 35 obtained from the NCAR Sabreliner at short distances, 50 to 1600 m behind the source aircraft. The data were obtained during the Contrail Studies Airborne Exhaust (CSAE) project near the Front Range of the Rocky





Mountains in Colorado and Wyoming, including 18 individual flights (within 40-44°N, 102-107°W) in March/April 1989 (Baumgardner et al., 1998). The raw data and some videos are available online (http://data.eol.ucar.edu/codiac/dss/id=252.004). Six flights, between 23 March and 13 April 1989 (see Supplement), were used to derive the data. From these, the mean values of flight conditions when the Sabreliner was in the Learjet exhaust

plume, based on condensation nuclei (CN) concentrations, can be determined: p=256±14 hPa, pressure altitude = 10.22±0.36 km, T=-55.3°C±2.6°C, RHi= 104±21%, TAS= 189±1.5 m s$^{-1}$. Unfortunately, the Learjet positions versus time are not in the archive files, so that the distances between the two aircraft cannot be reconstructed. Instead we complete the data using published information. Some details from these measurements are given in Andronache and Chameides (1998), who compared the observations to model results. The measured exhaust plume was compared with a wake-flow model and found

to stay within about W = 30 (25-40) m during the first 25 s plume age (Baumgardner et al., 1998). The particles were sized and counted with the then new PMS Forward Scattering Spectrometer Probe FSSP-300 optical particle counter, detecting particles in the size range from 0.35 to 20 μm (Baumgardner et al., 1992). Most of the ice particles had sizes between 0.35 and 1.5 μm, with 10 μm maximum size. The ice particle concentration reached a maximum of about 7000 cm$^{-3}$ at plume ages of about 3.5 s. For ages > 3.5 s, the observed concentrations are consistent with model studies (Andronache and Chameides,

1998; Kärcher and Yu, 2009). For shorter ages, the measured concentrations are far smaller than the model results. The ice-particle concentration at short distances might be low because the sampling was mainly outside the core of the younger contrails or because contrail particles had not yet grown to detectable sizes. Moreover, particle concentrations of 7000 cm$^{-3}$ are near the upper detection limit of the FSSP-300. The IWC values estimated from the reported particle sizes are about 10 to 50 mg m$^{-3}$ at maximum, roughly as simulated (Andronache and Chameides, 1997).

**3.1.3 ICE-1989: Contrails in cirrus**

The properties of aged contrails from airliners were derived from observations during the International Cirrus Experiment (ICE) 1989 over the German Bay of the North Sea (Raschke et al., 1990). Here, we report results from three ICE days (experiment numbers ICE-206, -210, -216 (Hennings et al., 1990)). The source aircraft and traffic details are unknown and contrail ages are estimates.

A contrail of 300-480 s age, produced by an airliner en route from Schiphol, Netherlands, was observed about 40 km east of Helgoland with the DLR Falcon aircraft during flight ICE 206, 24 September 1989. The observations were first presented by F. Albers et al. at a conference (1990, unpublished). A photo, microphysics data, and turbulence measurements for this case are reported in Quante (2006). The Falcon was equipped with PMS instruments (FSSP-100 and OAP-2D-C) operated by the Research Center at Geesthacht, Germany (formerly GKSS) (Gayet et al., 1993). Some microphysical

properties are given in comparison to remote-sensing and simulation results by Betancor Gothe and Graßl (1993) and Boin and Levkov (1994).





The measurements were performed along a 20-km long distance essentially in the center of the contrail. Kinetic energy spectra and rms values of the three velocity components show considerable turbulence inside the contrail (order 0.18 m s$^{-1}$, quasi isotropic), significantly larger than in the ambient cirrus (0.03 m s$^{-1}$, stronger horizontal than vertical). The PMS FSSP-100 and OAP-2D-C probes detected ice particles larger than 3 μm with concentrations of 0.1 cm$^{-3}$ and IWC of ~40 mg m$^{-3}$.

The Helgoland radiosonde indicated near-ice saturation at 8.5-10 km altitude (Hennings et al., 1990).

Higher ice particle concentrations (up to 1.5 cm$^{-3}$) were derived for another contrail, observed over the North Sea during ICE 210 on 13 October 1989. In-situ measurements were obtained with a PMS FSSP-100 and a PMS 2D-C probe onboard the MERLIN aircraft of the Centre d'Aviation Météorologique, Meteo France (Gayet et al., 1993), and with an upward looking lidar ALEX-F (Mörl et al., 1981) on a DO-228 aircraft operated by DLR (Gayet et al., 1996a). Gayet et al. (1996b)

analyzed both the FSSP-100 and 2D-C probes and listed data for $n_{ice}$, $r_{eff}$ and IWC in addition to T and altitude for four contrail samples. The MERLIN flew a 120-km long racetrack pattern about 5 to 10 km east (downwind) of the upper air traffic route UA7, between about 54°N, 7.3°E and 55°N, 7.8°E, mostly at 7000-7900 m altitude, between 11:15 and 13:05 UTC. The air was ice-supersaturated between 6200 and 8200 m, and warmer than expected from the SAC criterion. Cirrus was observed in the southern part of this flight. The contrail age 600 (400-700) s was roughly estimated from the distance to

the air traffic route, given wind speed and wind direction. Remote sensing observations for an additional ICE case are reported below.

### 3.1.4 FIRE/ARM: Multiple contrail data from self-encounter

Poellot et al. (1999) reported microphysical characteristics of 21 jet contrails from in-situ measurements and discussed radiative properties of the contrails. For 12 cases, the observed contrail was generated by the sampling aircraft, the

University of North Dakota (UND) Cessna Citation 500, at ages of 187-2640 s after generation. The sources and ages of the contrails in the other cases are unknown and not used here. Ice particles of about 1 to 50 μm were measured with the FSSP-100 particle spectrometer, and larger ice particles with 2 D-C and 1 D-C PMS OAPs. For each case, the paper lists T, p, mean and maximum $n_{ice}$ concentrations, average sizes, and CN concentrations indicating exhaust contributions. The CN concentrations do not correlate well with plume ages, likely because of sampling at variable distances from the plume center.

Data used from this study include $n_{ice}$, $r_{vol}$, IWC, and W for the Cessna Citation contrails, see Figure 1. Size spectra have peaks between 5 and 8 μm diameters. Smaller ice particles may be underrepresented due to the lower size limit of the FSSP-100.

### 3.1.5 SULFUR: Fuel sulfur influence on contrails

SULFUR 1 was an ad-hoc project of DLR, with a single flight by a propeller aircraft for visual contrail observations behind

the DLR ATTAS, a twin-engine jet aircraft used as a test facility. It burned different fuels on the two engines, with known fuel sulfur content (FSC, 2 and 250 μg g$^{-1}$), in North Germany, 13 December 1994 (Busen and Schumann, 1995). The





changes in FSC were found to have minimal impact on contrail onset, visible about 25-35 m behind the engines at threshold conditions. The jet contrails had diameters of about 2 m each and this value is used to estimate the initial width/depth and cross-section area. The data showed that the SAC threshold temperature is slightly higher for finite overall propulsion efficiency. These observations were of importance in understanding the impact of fuel sulfur on aerosol and contrail

formation, and were used for comparisons to model studies in several follow-on papers (Kärcher et al., 1996; Schumann, 1996; Brown et al., 1997; Andronache and Chameides, 1998; Kärcher et al., 2015).

SULFUR 2 continued the investigation of sulfur impact on aerosol and contrails using airborne in-situ measurements from the DLR Falcon behind the ATTAS aircraft for fuels having different FSC (170 and 5500 µg g$^{-1}$). The single-flight experiment was performed over Southern Germany, 22 March 1995, for temperatures at and below the contrail threshold.

Here, we include the W and D data at about 3 km distance (18-s age) behind the aircraft. The measured $n_{ice}$ = 70 cm$^{-3}$, obtained with an FSSP-100, with a lower size threshold of 3 µm, underestimates the true concentration of very small ice particles by possibly an order of magnitude and is, therefore, not plotted in Figure 1. Again, the observations influenced aerosol and contrail formation modeling (Gierens and Schumann, 1996; Kärcher et al., 1998; Yu and Turco, 1998).

SULFUR-3 performed exhaust measurements at the surface, not discussed here. SULFUR-4 sampled the contrail of the

ATTAS and an Airbus A310 with improved aerosol and cirrus particle measurement instrumentation, including the FSSP-300, and a prototype Multi-angle Aerosol Spectrometer Probe MASP (size range 0.4 to 10 µm) (Baumgardner et al., 1996) with simultaneous measurements of particle forward/backward scattering (Petzold et al., 1997; Kuhn et al., 1998). The results include a total of 6 data points, from sampling for low and high FSC (6 and 2700 ppm), in the plume center and at the plume edge, for young plumes (2-s old) and one case for a 10-s age behind the ATTAS. One further 5-s age A310 data point

is included in Schröder et al. (2000).

Among others, the results show strong variations in the particle size distributions from the plume center to the diluted plume edge for young contrails. The respective spectra were identified by the excess temperature above ambient air (>1.5 K in the plume center and <0.5 K at the plume edge). In the plume center, the contrail particle concentrations were of comparable magnitude for both high and low FSC. The $r_{eff}$ and IWC values are lower by factors 1.5 and 3.5, respectively, in

the high sulfur fuel case compared to the low sulfur fuel case. Schumann et al. (2002) provide a summary of the SULFUR 1-7 projects, with experimental details and results of the aerosol properties for variable fuel-sulfur contents.

### 3.1.6 AEROCONTRAIL: Contrail cirrus transition

Schröder et al. (2000) discuss a series of airborne in-situ measurements of microphysical properties of contrails and cirrus clouds from more than 15 airborne missions over central Europe, and described the transition of the ice particle size

spectrum from fresh contrails to contrail-cirrus. The measurements were performed as part of various national and European projects ("Schadstoffe in der Luftfahrt" (SiL), i.e., "Pollution from Aviation"; DLR-CIRRUS and SULFUR, and the European project AEROCONTRAIL), between March 1996 and April 1997. Concentrations and sizes of the smallest ice





particles were measured with a PMS FSSP-300. Other optical spectrometers, a Hallett-type replicator, and a novel polar nephelometer (Gayet et al., 1998) were used to characterize the size, concentration, shape and scattering phase function of larger ice particles. The paper lists the mean properties of the contrails and cirrus clouds measured in terms of temperature, $n_{ice}$, $r_{vol}$ and $r_{eff}$ (actually diameters were reported). The given humidity values mostly suggest subsaturation, but the large

IWC detected in the contrails suggests humidity exceeding ice saturation. The pressure values were obtained from Schumann (2012). COLI includes information and a few additional data from the campaigns CIRRUS'92 and CIRRUS'94 (Strauss et al., 1997) and from an unpublished project report (Wendling et al., 1997). For most of the contrails, the aircraft causing the contrail and the corresponding contrail ages were identified from direct evidence or from air-traffic-controller information.

For cases A1 and U of Schröder et al. (2000) (60 and 1200 s aged contrails behind an Airbus A319 and an undefined

aircraft), scattering phase functions as measured by the Polar Nephelometer, or as derived from FSSP-100 data with assumptions on the particle habit, were presented in Gayet et al. (1998). The scattering phase function analyses showed that the young contrail particles had quasi-spherical shapes. It was noted that the older contrail and ambient cirrus particles were far more irregular, possibly with rough surfaces.

### 3.1.7 SUCCESS: Transition of a young persistent contrail into a precipitation trail

The Subsonic Aircraft Contrail and Cloud Effects Special Study (SUCCESS) was a multi-aircraft field campaign in April and May 1996 over the Central/Western USA (Toon and Miake-Lye, 1998). The NASA DC-8 aircraft sampled its own contrail at ages of up to 0.75 h during a flight over the Pacific Ocean, off the Northern Californian Coast, 12 May, 1996. The observations are discussed in several papers (Heymsfield et al., 1998; Jensen et al., 1998a; Lawson et al., 1998; Minnis et al., 1998; Tan et al., 1998; Twohy and Gandrud, 1998). The investigations showed the transition of a young persistent contrail

into a precipitation trail. The contrail was generated at about 10.6 km altitude (p = 239 hPa), in clear but highly ice-supersaturated (RHI < 160%) air; however, patchy cirrus was also observed. Ambient shear was highly variable, $Sh_T \sim 0.01$ $s^{-1}$ on average, with low Brunt-Väisälä frequency, $N_{BV} = 0.0066$ $s^{-1}$, suggesting strong mixing. The many small ice particles in the contrail core (diameters ~ 1 to 10 μm; volume concentrations ~ 10 to 100 $cm^{-3}$) reduced the vapor density to ice saturation. Along the contrail periphery, crystals grew to sizes > 300 μm. The particles were falling from the contrail into the

supersaturated environment below, feeding precipitation trails (Heymsfield et al., 1998). The larger ice crystals in the periphery were mostly columns and bullet rosettes, with featureless shapes as deduced from the scattering phase functions (Lawson et al., 1998).

Data derived from these studies included in Figure 1 are concentrations $n_{ice}$ =100 (50-150) $cm^{-3}$, IWC = 5 (<20) mg $m^{-3}$ for the contrail core, and a contrail depth of 500 (400-600) m. An age of 2000 (1300-2600) s was reanalyzed from the

original DC-8 data. The TAS was 232 m $s^{-1}$ (Twohy and Gandrud, 1998). The wind speed components at contrail level were about 30 and 20 m $s^{-1}$ in easterly and northerly directions, respectively. Some of the high particle concentrations may have resulted from two DC-8 flight paths long the same flight track.



Simulations of this contrail showed that the contrail spread by shear and possibly radiative heating driving local updrafts, and the $n_{ice}$ observed could be explained with ice crystals nucleated during the initial contrail formation (Jensen et al., 1998a). Further results from remote sensing are given below.

### 3.1.8 FIRE-SUCCESS: Contrails from airliners

On another occasion, during SUCCESS in connection with the First International Satellite Cloud Climatology Project Regional Experiment (FIRE), a flight took place over northern Oklahoma and south-central Kansas. The air was mostly cloud-free; contrails were present but not persistent. The DC-8 followed a B757 at distances ranging from 3 to 20 km (Tan et al., 1998). At 18:54 UTC 4 May 1996, Baumgardner and Gandrud (1998) performed in-situ measurements with the DC-8 in the contrail about 5 km behind a B757 over south-central Kansas, for ages of about 30 (25-35) s, at -63°C, 197 hPa, with ~40

ppm of water vapor molar mixing ratio (about 112% RHi). The measurements, performed with the MASP instrument, yielded crystal concentration $n_{ice} \approx 100$ (50-200) cm$^{-3}$ and volume mean radius $r_{vol} \approx 2$ (1-3) μm, as determined from the published plots. For $n_{ice} = 100$ cm$^{-3}$ and $r_{vol} = 2$ μm, the computed IWC is 1.3 mg m$^{-3}$. Fig. 2 in Baumgardner and Gandrud (1998) shows IWC of 20-80 mg m$^{-3}$, likely including contributions from ambient cirrus; these values are not included in the Figure 1.

Data from an earlier part of the same flight behind the B757 at 18:12-18:14 UTC, at a slightly lower altitude (p = 203.5 hPa, T = -61°C) and lower ambient humidity (80% over ice), are discussed in other papers (Jensen et al., 1998c; Twohy and Gandrud, 1998) including data on crystal habits which are important because these have a large effect on the radiative properties of contrails (Goodman et al., 1998; Markowicz and Witek, 2011). Here, the MASP data are cited with $n_{ice} = 1$-50 cm$^{-3}$ for crystals >1.5 μm in radius, and the simulation considers a 70-s aged contrail (Jensen et al., 1998c). The measured

concentration appears low for this rather uncertain age. Hence, these data are excluded from the plots.

### 3.1.9 CRYSTAL-FACE and CR-AVE: Low temperature contrails at the tropical tropopause

During Cirrus Regional Study of Tropical Anvils and Cirrus Layers-Florida Area (CRYSTAL-FACE), a field experiment addressing cirrus from deep convection, measurements were made in the upper tropical troposphere on board the NASA WB-57F while sampling its own contrail at 300 to 2400 s age near South Florida (25.8°N, 81°W) on 13 July 2002. The data

are available from https://espoarchive.nasa.gov/archive/browse/crystalf/WB57/20020713. At the time of contrail formation (about 64700 s UTC) the aircraft was descending with a mass of about (24300±450) kg and fuel consumption of 680 kg h$^{-1}$, based on information from the aircraft operators. The contrail was observed within some thin cirrus having large ice particles (Jensen et al., 2005). Gao et al. (2006) report measurements of water vapor and nitric oxide (NO) molar mixing ratio, p, T, RHi, ice and particle number, size, surface area, and volume as derived from measured 1-s size distributions. The contrail

was formed at low temperatures (-77°C at 120 hPa, ~15 km altitude). The temperature data agree well with the nearby Miami radiosonde data. From the plotted results we read mean values for 780 s and 1560 s contrail ages with $n_{ice}$ ~50 to 160



cm$^{-3}$ and low IWC of 0.18 to 0.23 mg m$^{-3}$, implying low $r_{vol}$ values of 0.7 to 1 µm, and $r_{eff}$ of 1 to 2 µm. Large, steady-state RHi values were measured inside the contrails. The analysis suggests that nitric acid increases relative humidity in low-temperature cirrus clouds (Gao et al., 2004; Gao et al., 2016).

The same measurements were analyzed independently with respect to microphysical properties in a Master's thesis (Mullins, 2006) (Timothy Garrett, advisor). The thesis presents reevaluated CAPS (Baumgardner et al., 2001) data implying smaller extinction ($\beta_{ext}$) values. Together with a larger IWC estimate, including a few larger ice particles possibly from ambient subvisible cirrus, nearly a factor 2 smaller effective radius $r_{eff}$ is derived from the IWC and $\beta_{ext}$ values. The contrail W is estimated (about 200 to 1500 m from the first penetration to 1800 s age) from a sequence of photos at various times of contrail development. Using a wake vortex descent model, the total depth D of the contrail was estimated. The reported W is consistent with age, ambient wind shear, and the estimated D. The effective depth $D_{eff}$ is, however, likely far smaller than D because of shear induced cross-section deformation, giving smaller cross-section $A_c = D_{eff} W < D W$ and, hence, about constant total number of ice particles $N_{ice}$. The data suggest quite high ice particle numbers per fuel consumption.

At ~10 K lower temperatures, a contrail was observed during the NASA Costa Rica Aura Validation Experiment (CR-AVE) over the Pacific south of Costa Rica on 1 February 2006. During validation flights for satellite retrievals, airborne measurements were performed again with the NASA WB-57F, sampling its own contrail in the uppermost tropical troposphere. The contrail observations are described briefly in two informal conference papers (Flores et al., 2006; Lawson et al., 2006). The cirrus data are discussed in Lawson et al. (2008). Here we compile the essential data.

Raw data are available from https://espoarchive.nasa.gov/archive/browse/cr_ave/WB57/20060201. Only 30-s averaged CAPS data and no CO and HNO$_3$ trace gas data are available for the contrail period. At the time of contrail formation (about 64700 s UTC), the aircraft was cruising at a constant altitude of 18 km (79 hPa) near the equator at 83°W. It measured its own contrail between 65350 and 65700 s UTC. Fuel consumption and aircraft mass were estimated by the flight operators as 1270 kg h$^{-1}$ and (24800±450) kg. The contrail formed at a low ambient temperature, near -88°C, consistent with radiosonde data of San Andres Island (12.58°N, 81.71°W) from 12 UTC the same day. The contrail was penetrated at least four times, at plume ages of 410 to 1150 s. The contrail penetration times and positions were computed for each flight path with the measured wind data. The atmosphere was stably stratified ($N_{BV}$ = 0.02 to 0.03 s$^{-1}$) with variable shear around $Sh_T \approx 0.02$ s$^{-1}$.

Our analysis shows that the penetrations occurred mainly above the core contrail, in the secondary wake. For given wake scales $b_0$ = 29.3 m, $t_0$ = 19.0 s, $w_0$ = 1.54 m s$^{-1}$, the product $N_{BV} t_0$ controlling wake descent in stratified air is about 0.47, implying a wake vortex descent of about 6 $b_0$ =178 m (Schumann and Heymsfield, 2016). During contrail measurements, the flight path was higher than at the times of contrail formation. Hence, the contrail would not have been penetrated at altitudes as measured, had there been no upward motion in the ambient air, of about 0.13±0.05 m s$^{-1}$.

The maximum ice concentration was found in the last penetration, either because it occurred closest to the contrail core or because the contrail ice particles had grown in ice supersaturated air to sizes detectable by the CAPS instrument (0.5 µm).





The contrail penetration is also reflected in weak and noisy changes of some of the wind speed, temperature, $CO_2$, water vapor, and condensation nuclei measurement data, which are of comparable magnitude in all four penetrations. The relative humidity is uncertain; the four available water vapor measurements differ considerably (Lawson et al., 2008). The water data from WB-57's NOAA frost point measurements show the lowest values with subsaturation outside the contrail and high

supersaturation inside the contrail, but apparently with some timing uncertainty. The $H_2O$ peaks fit the other peaks better when shifted by 60 s forward in time. The peaks agree with expected $CO_2$ and $H_2O$ molar mixing ratio increases $\Delta c = (29/44)\,EI_{CO2}/N_{dil}$ and $(29/18)\,EI_{H2O}/N_{dil}$ of order 1 ppm for the given plume ages and dilution as in Eq. (1). For a fuel flow per flight distance of $m_F = 2$ g m$^{-1}$, this implies a cross-sectional area $A_c = m_F/(\rho\,N_{dil})$ of about 0.025 km$^2$ (e.g., W = 500 m and $D_{eff}$ = 50 m).

The CAPS instrument shows low concentrations of rather large ice particles in the ambient air and, as the contrail was penetrated, clear concentration peaks, mainly of small ice particles (90% of the particles were smaller than 2 μm diameter, maximum diameter 25 μm). Concentrations at 1-s time resolution (Flores et al., 2006) reach maximum values of 250 cm$^{-3}$. As derived from the forward to backscatter ratio from the CAS (one component of the CAPS), approximately 40% of the particles have a refractive index between 1.35-1.40, suggesting the presence of sulfuric acid. The habits were 15% spherical

and 50% with aspect ratios between 1.3 and 1.5 (also derived from the CAS forward to backscatter ratios). The spatial distributions of particles in the contrail are non-Poissonian and suggest smaller-scale contrail cloud structures with unknown origin. From the 30-s mean values available in the database (65679 to 65709 s), the contrail contributes $n_{ice}$ of 0.51 cm$^{-3}$ in the first channel, a total of 0.78 (<2.5) cm$^{-3}$ in all channels, IWC of 0.20 mg m$^{-3}$ ($r_{vol}$ = 4 μm), and $\beta_{ext}$ of 0.11 km$^{-1}$ ($r_{eff}$ = 3 μm).

Particle size distributions and images of the contrail ice particles from the measurements in the contrail were shown in comparison to subvisible cirrus (SVC) within about 1 km of the tropopause during this mission (Lawson et al., 2008). In the SVC, the average ice particle number concentration (0.066 cm$^{-3}$), extinction coefficient (0.009 km$^{-1}$), and IWC (0.055 mg m$^{-3}$) were far lower than in the contrail.

### 3.1.10 PAZI-2: Contrail optical particle properties

As part of the DLR project Particles and Cirrus-2 (PAZI-2), the Falcon was used to measure a persistent contrail formed by an Embraer E170 mid-sized jet aircraft (Febvre et al., 2009). The contrail formed near Berlin, Germany, at about 11900 m altitude, at -60°C, and at RHi of 108 - 131%, > 8 K below the contrail threshold temperature. Mean ambient shear perpendicular to the flight direction, derived from in-situ wind measurements, varied within Sh = 0.005 -0.01 s$^{-1}$, with local thin shear-layers, and $N_{BV}$ ~0.011 s$^{-1}$. The contrail was penetrated several times at two evolving stages, first at about 150 s

contrail age, just after the vortex phase, and later for 660 s and 1200 s contrail ages, in the dispersion phase. The D and W evolved from 120 m and 350 m estimated for the young contrail to be about 150 m and 1800 m in the aged contrail. Many of the later measurements were likely taken in the secondary wake above the sinking primary wake core. The number of ice





particles larger than 1 μm from FSSP-300 data was 68.3/18.3 cm$^{-3}$ for the young/aged contrail regions. The IWC was low, near 1 mg m$^{-3}$. Optical extinction was measured with a polar nephelometer (Gayet et al., 1998) at 804-nm wavelength, decreasing with contrail age from 0.48 to 0.29 km$^{-1}$. Particles smaller than 10 μm contribute about 80% of the $\beta_{ext}$ and therefore dominate the optical properties. From IWC and $\beta_{ext}$, the effective radius was derived to yield 3 μm and 5.5 μm for

the young and the aged contrail. The asymmetry parameter g derived from the nephelometer data decreased from 0.827 to 0.787 with age, revealing quasi-spherical ice particles in the young contrail and more aspherical crystals later. Nearby thin frontal-cirrus contained larger and more aspherical crystals with $g \approx 0.7$, $r_{vol} > 20$ μm.

### 3.1.11 SCOUT-O3: Contrails and cirrus above tropical convective clouds

Contrail observations at low temperatures were derived from past Geophysica measurements. The Russian M-55 Geophysica

is a reconnaissance aircraft converted into an atmospheric research aircraft equipped with a large set of in-situ and remote sensing instruments, similar to NASA's ER-2 and WB-57. The M-55 reaches maximum pressure altitudes of 20 km. Measurements in the tropical stratosphere within and above clouds were performed with the M-55 as part of various European projects (Stefanutti et al., 2004; Corti et al., 2008; Vaughan et al., 2008; Brunner et al., 2009; de Reus et al., 2009; Weigel et al., 2009; Cairo et al., 2010). We found several indications for self-induced contrails for stratospheric conditions

(Schumann and co-authors, 2016). Measurement periods with Geophysica exhaust and contrail contributions were found near and above deep tropical convection in the tropics, during the project Stratospheric-Climate Links with Emphasis on the Upper Troposphere and Lower Stratosphere (SCOUT-O3) flights over the Tiwi Islands (131°E, 11.5°S), near Darwin, Australia in November 2005. The measurements were performed with the M-55 in coordinated flights with other research aircraft, e.g., the DLR Falcon, carrying an upward-looking lidar (Kiemle et al., 2008; Wirth et al., 2009). The analysis

provides remote sensing results based on photography, lidar, and NOAA Advanced Very High Resolution Radiometer (AVHRR) satellite data as explained in Section 3.2.11.

From in-situ measurements during three flights of SCOUT-O3 near Darwin (November 29, and 30 morning and afternoon, 2005), several potential contrail self-encounters in cirrus were identified from trajectory analyses. Potential encounters are those for which the flight path crosses computed plume positions within ±100 m vertical distance. The mean

measured cirrus properties at these times were taken as being representative for contrails at those positions. Since there is no proof that these cirrus clouds were actual contrails, these results should be interpreted with caution. We include these data in COLI with this annotation.

During the "Golden day" of the SCOUT-O3 project, 30 November 2005, six ice events longer than 30-s measurement duration were found in the morning flight above a deep convective cloud, "Hector", over the Tiwi Islands near Darwin, at

altitudes up to 18.7 km, in the lower stratosphere, as documented in detail by de Reus et al. (2009). Several indications suggest that these ice events were influenced by nonvolatile aircraft aerosol and contrails (Schumann and co-authors, 2016). COLI includes the measured data with trajectory-based age estimates. The SCOUT-O3 and CR-AVE results are essential in





extending the set of observations to lower temperatures. The conclusions of this study do not change significantly when the potential-contrail data from SCOUT-O3 are omitted (see Section 4).

### 3.1.12 CONCERT-2008: Aircraft influence on contrails

The Contrail and Cirrus Experiment (CONCERT) determined aircraft-dependent contrail properties (Voigt et al., 2010).
Line-shaped contrails were measured with the Falcon for contrail ages of several minutes. Contrails from different aircraft were probed in the upper troposphere at temperatures of -57 to -51°C. Here, we include results for 4 airliners of different types (CJ-2, A319, A340, A380), all in near ice-saturated conditions. Particle sizes, $\beta_{ext}$ and IWC were measured in the size range 0.45–17.7 μm with an FSSP-300, and the scattering phase function and $\beta_{ext}$ with a polar nephelometer. OAPs detected larger ice particles. In addition, various chemical species were measured ($SO_2$, NO, $NO_y$, $HNO_3$, HONO, HCl, $O_3$ and CO)
(Jurkat et al., 2011). Values of $\beta_{ext}$, τ, and IWC are given in Voigt et al. (2011). The aircraft impact on contrails was discussed for comparable ambient conditions for the A319, A340 and A380 aircraft by Jeßberger et al. (2013). Jeßberger et al. (2013) and Unterstrasser (2014) compare the measured results with model simulations. Schumann et al. (2013b) relate the measured particle concentrations to dilution (from wake vortex scales and trace species concentrations) and estimate soot and ice number emissions. Gayet et al. (2012a) find increasing asphericity of contrail particles during ageing in the A380
contrail for plumes as old as 250 s.

Some of the contrails were embedded in cirrus with low concentrations of large ice particles (Kübbeler et al., 2011). The RHi observed outside the contrails was often slightly below ice saturation. Most contrails were optically thick, as documented in photos and data (Voigt et al., 2011), and the IWC values were far higher than explainable by the engine water emissions in subsaturated air. Hence, the ambient air must have been ice-supersaturated during the time of contrail IWC
growth (Voigt et al., 2011; Gayet et al., 2012b; Jeßberger et al., 2013).

### 3.1.13 CONCERT-2011: Ice saturation inside contrails

CONCERT 2011 was an extension of CONCERT (Voigt et al., 2014). Again several contrails were probed with the Falcon. Kaufmann et al. (2014) showed that the measured humidity was fully consistent during these measurements, with the expected ice saturation inside the contrail. From these measurements, three contrails are included in the COLI set for contrail
ages of 80-160 s. The A319 case discussed by Kaufmann is exceptional in showing a contrail with many small ice particles ($n_{ice} = 117$ cm$^{-3}$, $r_{vol} = 0.5$ μm) and consequently low ice water content (IWC = 0.06 mg m$^{-3}$). Two further cases are included, a B77W (freight version of a B777) flight during 16 September 2011 with FSSP-300 and polar nephelometer (PN) data, and another B77W flight on 24 September 2011 (PN only). The contrail ages (80 - 160 s) were determined from air traffic data and with radiosonde and Falcon-derived wind data. The measured high NO molar mixing ratios, exceeding 10 nmol mol$^{-1}$,
clearly identified the data as in-plume measurements.





### 3.1.14 COSIC: Self-induced spiral contrail cloud sampled with improved optical size spectrometers

Jones et al. (2012) present results from sampling a self-induced "spiral-contrail" over England and the North Sea using the FAAM BAe146 UK research aircraft as part of the Contrails Spreading Into Cirrus (COSIC) study. Similar spiral type contrails have been observed for reconnaissance aircraft (Schumann, 2002; Haywood et al., 2009). In contrast to the racetrack pattern flown by the DC-8 during SUCCESS, in which the DC-8 followed the contrail position moving with the wind (Lagrangian profile), here the spiral results from flying in an orbit over the same ground position while wind blows the contrails downwind (Eulerian profile). The flight, named B587, took place in an ice-supersaturated area between ~14 and 19 UTC 19 March 2011. Several parts of the BAe146 contrail were sampled at contrail ages of 420 to 1800 s. Contrails from other aircraft were also sampled, but for unknown ages. The microphysical properties were measured with newer optical size spectrometers (CAS-DPOL, 0.6 - 50 μm and CIP-GS, 15-900 μm). The BAe146 is propelled by four turbofan engines. It would be interesting to know whether aircraft with such engines emit the same number of soot particles as comparable jet aircraft.

The contrails were observed with remote sensing up to 2400 s age. The persistent contrails were found to consist of small (~10 μm) plate-like crystals where growth of ice crystals to larger sizes (~100 μm) was typically detected when higher water vapor levels were present. From the in-situ data, $\beta_{ext}$ values were calculated and found to be 0.01 to 1 km$^{-1}$. The high values may apply to fresh contrails and the low to aged contrails. Limited water vapor supply was thought to have suppressed ice crystal growth. A table in Jones et al. (2012) lists individual values of $n_{ice}$ from the two instruments for contrail ages of 80 to 1800 s, together with T and RHi values, which we include into the COLI table, but no IWC and $\beta_{ext}$ values are provided for the specific contrail events.

### 3.1.15 ML-CIRRUS: Aged contrails and cirrus in the North Atlantic traffic corridor

Many aged contrails were sampled on the High Altitude and Long Range Research Aircraft (HALO) during the Mid-Latitude Cirrus (ML-CIRRUS) campaign in 2014 (Voigt et al., 2016). One specific example is reported here. The contrail was identified from enhanced NOx emissions (ΔNOy = 40 pmol mol$^{-1}$) at p = 215 hPa, T = -62°C, RHi = 102%, along a 18-s long segment at 36150 s UTC 26 March 2014 during a flight at FL 370 crossing the North Atlantic flight corridor in a southbound direction at 14.0°W 53.7°N, west of Shannon. For a TAS of 220 m s$^{-1}$, the segment length corresponds to W = 4000 m if perpendicular to the contrail direction. The measured wind components are (6, 20) m s$^{-1}$ in (east, south) directions. The contrail was found to have still higher concentrations of relatively small ice particles than ambient cirrus. The CAS data imply an IWC of 0.9 mg m$^{-3}$, and $r_{eff}$ = 9.5 μm and $r_{vol}$ = 6.5 μm. Due to the presence of larger particles (>50 μm) in the contrail cirrus, these data can be interpreted as lower limits. The $\beta_{ext}$ derived from the planar particle cross section of CAS and CIP data is 0.13 km$^{-1}$, significantly larger than the $\beta_{ext}$ from the surrounding cirrus of 0.07 km$^{-1}$, mainly due to the presence of higher number densities of small particles.



Based on radar-observed traffic data from EUROCONTROL for the UK air space, and from trajectory analysis using ECMWF wind data, one finds two possible source aircraft for the measured contrail. The observed contrail may have been generated by a B772 flying along a route from North America to Europe at FL 380, near 14.37°W, 55.77°N, at 29627 s (8:14 h) UTC, 1.8 h (6533 s) before the measurements. At the measurement time, the contrail from this flight crosses the HALO

flight path nearly perpendicularly. This conclusion derives from analyzing the nominal ECMWF winds and also when the horizontal wind speed components are increased by 2 m s$^{-1}$, which yields a better match between the ECMWF and in-situ measured winds. The ECMWF RHi values exceed 90% along the whole trajectory. The fact that the contrail was measured 1000 feet (300 m) below the flight path of the B772 is explained by some subsidence (as indicated by the ECMWF data) and the contrail descent; but this makes an alternative explanation more likely: Alternatively, the contrail could have been

generated by a Falcon-900 (F900) aircraft on a similar eastbound route, but at FL 370, near 18.61°W, 55.72°N, at 10380 s (2:53 h) UTC, 7.16 h (25780 s) before the measurements. The trajectory analysis for this alternative contrail matches the HALO position only for 2 m s$^{-1}$ enhanced wind speeds and the computed contrail cuts the HALO flight path at an angle of about 45° (implying W = 2800 m). Here, the ECMWF RHi along the trajectory was >97%, and possibly the uppermost parts of the F900 contrail (generated at the same flight level) were measured.

For the given ages and an estimated emission index $EI_{NOx}$ = 20 g kg$^{-1}$, one would expect NOy increases of 1600 pmol mol$^{-1}$ (B772) or 533 pmol mol$^{-1}$ (F900), if the dilution followed the dilution law, Eq. (1). The computed concentrations are far higher than the measured 40 pmol mol$^{-1}$. An alternative dilution law (see below), $N_{dil}$ = 20 $(t/t_0)^{1.7}$, $t_0$ = 1 s, produces NOy increases of 206 pmol mol$^{-1}$ (B772) or 20 pmol mol$^{-1}$ (F900). Since the dilution laws are accurate at best to a factor of 3, both results are approximately consistent with the observations, so that the older F900 contrail appears to be more plausible.

Figure 1 shows that both results extend the set of available in-situ data to higher ages, and both are approximately consistent with CoCiP and other data.

### 3.2 Remote sensing

#### 3.2.1 CIAP: Contrails in the jet and wake vortex phases

During the Climatic Impact Assessment Program (CIAP), considering the impact of supersonic transport on the ozone layer

and on climate, the vertical and lateral dimensions of a B-59 jet aircraft contrail were determined from ground-based camera observations near San Francisco, likely in fall 1972. The contrail formed in the upper troposphere at 12.77 km and lasted for about 1800 s (Conti et al., 1973). The contrail scales were derived during the wake phase and until the early dispersion phase (about 600 s), with maximum W and D of 500/400 m (Hoshizaki et al., 1975). Sounding data and the day of measurements were not reported.



### 3.2.2 ICE-1989: Combined lidar and satellite observations of contrails

In addition to in-situ measurements during ICE (section 3.1.3), four contrails were observed by an airborne lidar during the ICE-216 flight over the German Bay of the North Sea on 18 October 1989. The contrails were oriented about parallel in the east-west direction and embedded in optically thin (0.01 to 0.1) cirrus. The aircraft flew a butterfly pattern near 7°E, 54°N

(Hennings et al., 1990). The upward looking backscatter lidar ALEX-F (1064 nm) was flown on a DO 228 at 3.55 km altitude with 100 m s$^{-1}$ TAS, southbound. The DLR Falcon, equipped with a Counterflow Virtual Impactor (CVI) and a PMS 2D-C, sampled in the cirrus above FL 300, and found cirrus particles, predominantly smaller than 50 μm, over Norderney between 13 and 14 UTC (Ansmann et al., 1993; Ström et al., 1994). The Helgoland radiosonde indicated ice saturation at 8-12 km. A combined Raman elastic-backscatter lidar observed the cirrus from Norderney (7°13'E, 53°43'N) and data were

reported for 10:45-16:30 UTC that day (Ansmann et al., 1992). The observations show a cirrus layer extending from 9 to 11.8 km altitude with τ ≈0.2 (0.1-0.4), extinction-to-backscatter ratios of 5 to 10 sr, and backscatter coefficients mostly at 0.1 km$^{-1}$ sr$^{-1}$, with 150-s peaks up to 0.6 km$^{-1}$ sr$^{-1}$, possibly because of contrails. Inside this cirrus, three contrails were observed by the ALEX-F lidar at 12:21, 12:23, and 12:26 UTC, nearly coincident with a NOAA satellite overpass at 7.3°E, 53.6°N at 12:26 UTC (Kästner et al., 1993).

Betancor Gothe and Graßl (1993) applied the split window technique (Inoue, 1985) and compared measured brightness temperature differences near 11 and 12 μm of the NOAA-11 AVHRR scene of this day over the North Sea with radiative transfer model results and deduced the $r_{eff}$ and τ of contrail cirrus over the North Sea. Linear parts of the cirrus contained smaller ice particles ($r_{eff}$ <25 μm) with larger τ (<0.2) compared to ambient cirrus, roughly consistent with the above mentioned lidar results.

A remarkably thick contrail was observed at low altitude below a thin cirrus deck, along the same flight near 6.8°E, 53.4°N, at 13:14 UTC (Schumann and Wendling, 1990). The value of τ was derived from lidar using a shadow technique (Ruppersberg and Renger, 1991) and from the radiances measured in five channels (2 visible, 3 infrared) of the NOAA-11 AVHRR satellite data. The contrails were observed with W = 1.2, 1.8, 3.0, and 3.0 km, within 8.8-9.1, 10.0-10.2, 8.1-8.5, and 7.6-8.3 km altitudes, with 200/300, 150/200, 250/400, 400/700 m mean/maximum D, at the clock times 12:21, 12:23,

12:26, and 13:14 UTC, respectively. The maximum values of τ (at 1064/530 nm) derived from the lidar/AVHRR data are 0.35/0.41, 0.36/0.30, 0.50/0.34, and 1/n.a. The mean τ may be 50-70% of these values, consistent with an analysis of AVHRR data for the same cases by Betancor Gothe and Graßl (1993). Nearby radiosondes at 13 UTC (Quante, 2006) reveal temperatures of -43,-51, -41, -37.4°C, a wind speed of 10 m s$^{-1}$ from 210°, weak shear, $N_{BV} \approx 0.007$ s$^{-1}$, and RHi up to 90% above 8 km.

The lifecycle of the cirrus clouds (up to 60 h) was observed with imagery of the Geostationary Satellite METEOSAT, radiosondes, and several ground-based lidar observations, indicating that cirrus particles have shorter lifetimes than cirrus clouds (Szantai et al., 2001). Fallstreaks and broadened contrails were observed during the whole day, starting from thin





cirrus at the tropopause (Ansmann et al., 1993). The large contrail widths in low-shear environment suggest large ages. The wind derived from the Falcon data was near 10 m s$^{-1}$ from 200° (about southwest) (Quante, 2006). Based on the distance (50-100 km) to upper air traffic air routes passing Emden (53.39°N, 7.24°E) in the east-west directions (and traffic data of 18 October 2008; those for 1989 are not available), and for given wind speed, we estimate the age as 1±0.5 h. During that same

day, cirrus was measured over Scotland (Francis et al., 1994) and an extensive contrail-cirrus deck was observed over Southern Germany (Schumann and Wendling, 1990) and the Northern Adria (Betancor Gothe and Graßl, 1993), to be discussed with other contrail cirrus cases. The original measurement data are no longer available.

### 3.2.3 SiL: Lidar measured wake contrails of a large commercial aircraft

Baumann et al. (1993) measured contrail properties using the airborne lidar ALEX-F on the Falcon and determined W and

descent speed of the contrail in the wake vortex behind a B747-200 cruising (TAS = 225 m s$^{-1}$) in clear air close to the tropopause near 49°N, 12°E (Munich, Germany) for ages of 19 to 49 s at 12:25 UTC 9 April 1991. The two counter-rotating vortices contained separate contrail lines 12-m wide and separated by 43 m laterally, implying W ≈ 55±4.7 m (data listed in Schumann (1994)). The optically effective depth measured from lidar backscatter plots was $D_{eff}$ ≈35±3 m, slightly decreasing with age because of sublimation in subsaturated air (Busen et al., 1994). The mean descent speed was 2.5 m s$^{-1}$

causing a total wake depth D of 125 m after 50 s. For the given flight level (350, p = 238 hPa), T = -63.3°C, ambient humidity (RHi = 39%), wind shear (0.002 s-1), and stratification ($N_{BV}$ = 0.027 s$^{-1}$) were derived from the nearby Munich radiosonde at that time. The SAC threshold is $T_{LC}$ = -50.4 °C for these values. Data on the Falcon flight path are available from the COLI data bank.

### 3.2.4 IFU-Lidar: Ground-based lidar observations of contrail geometry and optics

Freudenthaler et al. (1994) developed a ground-based lidar (at the Institut für Umweltforschung, IFU, Garmisch, Germany) for remote sensing of contrails. Contrails from commercial aircraft at cruise in the upper troposphere passing over Garmisch, Germany (47.5°N, 11°E) were observed between 1993 and 1996. The first paper describes a contrail, ~1 h old, moving with northerly winds roughly perpendicular to the flight path through the fixed vertically pointing lidar beam at 16 UTC 1 April 1993. (The given time of 17:00 may by 16 or 17 UTC.) The data imply: W ≈ 1350±50 m, D ≈ 250±20 m, $A_c$ = 0.164 (0.15-

0.18) km$^2$, τ = 0.15 (0.1-0.4), and $\beta_{ext}$ = 0.8 (0.16-1) km$^{-1}$. Data from the twice daily Schleissheim radiosonde (about 100 km north of Garmisch) were used to estimate T, p, $Sh_T$, and $N_{BV}$. The sounding reports RHi ~60%, but the contrail was growing in size, hence likely in ice supersaturated and possibly uplifting air near the Alpine foothills.

Freudenthaler et al. (1995) report measurements from a set of contrails observed with the scanning version of the same lidar for contrail ages of 60 to 3600 s and derived mean vertical and horizontal spread rate of contrails of 0.3 and 2.3 m s$^{-1}$,

respectively. The areal spread rate was 60 to 420 m$^2$ s$^{-1}$, mostly driven by shear. The ratio $A_c$/(W D) decreases from about




0.6 for the lowest width to 0.4 at 4-km width, and the lidar showed increasing shear-driven inclination of the contrail cross-sections with time.

Figure 2 shows that the width (~80 to 150 m) increases slowly for low ages < 200 s and then linearly with longer age. The vertical growth was often limited by the depth of the humid layers within the flight levels, as later found also in large

eddy simulations (Unterstrasser and Gierens, 2010a, b; Lewellen, 2014).

Freudenthaler et al. (1996) derived $\tau$ and depolarization values from the lidar observations. Data from this work in 1994 to 1996 (69 individual observations with contrail D, W, $A_c$, p, T and RHi, including $\tau$ for 48 observations) were made available for this paper. Those for 1995 and 1996 were partially reanalyzed for this purpose. Contrail age is provided for all of these data, but the aircraft types were not recorded. The data for 1994-1995 were obtained within the project SiL and a

German Science Foundation (DFG) funded project (Schumann, 1998), while those for 1996 are from the project AEROCONTRAIL (Ström and Ohlsson, 1998) and include observations reported by Sussmann (1999).

At T<-60°C, the observed depolarization values inside the contrails increase from about 0.1 to 0.5 with age, whereas for higher temperatures, T ≈-50°C, already the youngest contrails exhibit depolarization values of about 0.5, suggesting quicker formation of aspherical ice particles at higher temperatures. The $\tau$ results were compared with other results, e.g., by Kärcher

et al. (2009a).

Within AEROCONTRAIL, the contrail of a B747-400 aircraft, in rather turbulent ambient air in the upper troposphere, was measured with the IFU-lidar and from a ground based digital camera at life times of 5.7 to 50.3 s, together with turbulence measurements obtained on the Falcon (Sussmann, 1999; Sussmann and Gierens, 1999). The contrail was observed from the surface near Augsburg (48.3N, 10.7E), Germany at 12:37 UTC 22 April 1996. The contrail formed at p =

236.7 hPa and T = -52.3°C in a slightly unstably stratified atmosphere ($N_{BV}^2$ = -(3±2.4) $10^{-5}$ $s^{-2}$). A secondary wake was identified above the primary part of the wake in photos and lidar data. The humidity was not measured but the analysis suggests slightly ice supersaturated ambient air, possibly RHi = 101% (Sussmann and Gierens, 1999). The shear was highly variable, but low on average. The lidar identified cross-sectional areas for the primary and secondary wakes of about 4500±100 $m^2$ each, at 50 s contrail age. The contrail depth was D = 160 m and W = 90 m at this age. The $\tau$ derived from the

lidar data is 1.18 and 0.24 for the primary and secondary wakes, respectively. Apparently for another contrail, the primary-wake contrail sublimated at 170 s age and the contrail in the secondary wake was slightly inclined by shear with a depth D = 160 m and W = 100 m (Sussmann and Gierens, 1999). Nearly-simultaneous 100-Hz turbulence data reveal a dissipation rate of turbulent kinetic energy of (7.4±0.5)×$10^{-5}$ $m^2$ $s^{-3}$ and yield the rms values of horizontal (0.9 m $s^{-1}$) and vertical (0.5 m $s^{-1}$) turbulent velocity fluctuations. Follow-on studies, including numerical simulations of the contrails, showed that the ambient

humidity level and adiabatic warming of the primary wake during descent determines when a secondary wake is visible above a vortex pair and when it is not (Sussmann and Gierens, 1999, 2001).



### 3.2.5 SUCCESS: Long-lived racetrack contrail in satellite images

The contrail observed during SUCCESS on 12 May 1996 was generated while the DC-8 was flying a racetrack causing an oval contrail pattern. The oval was visible in geostationary satellite imagery (4-km resolution GOES-9) for seven hours (Jensen et al., 1998a; Lawson et al., 1998; Minnis et al., 1998). Additional data were derived from 1-km resolution polar orbiting NOAA-12 AVHRR data. The contrail visible $\tau$, the effective radius $r_{eff}$, and the temperature T of the contrails were derived from multispectral satellite data (Minnis et al., 1998). Data derived from these studies for contrail ages of 14400 (3600-25200) s in Figure 2 are W = 5 - 10 km, $\tau = 0.5\pm0.25$, and $r_{eff} = 30$ (15-40) μm. The study showed high variability of the results with time.

Besides the 12 May case, contrail clusters were observed with ages up to 17 h and $\tau$, on average, from 0.2 to 0.5 over the cloud lifetime. In all cases, cloud particle sizes increased as the contrails developed into cirrus clouds. The contrail clusters include several individual contrails and, hence, are not included in COLI.

### 3.2.6 SUCCESS-ER2: Remote sensing of microphysics and integral contrail properties

During the SUCCESS campaign, a multispectral thermal-infrared imaging spectroradiometer, an airborne Moderate Resolution Imaging Spectroradiometer (MODIS) simulator with about 100 m spatial resolution, and a 532-nm backscatter lidar (40 m and 7.5 m resolution in horizontal and vertical directions) (Spinhirne and Hart, 1990) were flown onboard the NASA ER-2 aircraft (Duda et al., 1998; Spinhirne et al., 1998). A young and an aged contrail were observed over northeastern Oklahoma 15:56 and 16:43 UTC 20 April 1996. The source aircraft and the contrail ages were not determined, but both contrails originated within 10 km of each other, and the aged contrail was observed 3000 s later than the younger one. In order to correlate the results with other data, we estimate contrail ages (900±300 and 3900±900 s). Pressure (302-335 hPa), temperature (-40 to -45°C), ambient humidity (near ice saturation), $N_{BV}$ (near 0.008 $s^{-1}$) and shear (low) are estimated from the Springfield, Missouri (83.37°W, 37.22°N) radiosonde observations at 0 UTC 21 Apr 1996, which is selected because it shows the highest humidity relative to the radiosondes in the neighborhood of northeastern Oklahoma.

This is the only study that determined the local and total ice water content (IWC and TWC) and the local and total number of ice particles ($n_{ice}$ and $N_{ice}$) in contrails per volume and per unit length by remote sensing. The method is complementary to the in-situ method of Knollenberg (1972). The method relates the IWC to $\beta_{ext}$ and $r_{eff}$ observable by remote sensing:

$$IWC = (2/3)\, \rho_{ice}\, \beta_{ext}\, r_{eff}, \text{ and } IWC = C^3\, n_{ice}\, (4/3)\, \pi\, \rho_{ice}\, r_{eff}^{3}\,, C = r_{vol}/r_{eff}. \qquad (3)$$

The values of $N_{ice} = n_{ice}\, A_c$, $\tau = \beta_{ext}\, D_{eff}$, EA = $\beta_{ext}\, A_c$, and TWC = IWC $A_c$ follow for the lidar-derived cross-section $A_c$ and related W and $D_{eff}$, where $A_c = W\, D_{eff}$. Spinhirne and Hart (1990) implicitly assumed C = 1, although they noted that $r_{vol}/r_{eff}$ may be as small as 0.5. Optical depth $\tau$, mean particle sizes $r_{eff}$, and temperature T of the contrail were derived by comparing measured brightness temperatures of contrails at several bands in the infrared window to radiative transfer simulations of the





clouds (Duda et al., 1998). The $\beta_{ext}$ profile is obtained from the analysis of the lidar data, assuming an extinction-to-backscatter ratio of 18 sr, giving a similar $\tau$. The integral of these variables observed over the contrail cross-section are used to evaluate $A_c$ and W.

The results show that the ice particles in the young contrail are generally smaller (7 μm) than those in the older one (20 μm), as expected, while the total number of ice particles $N_{ice}$ stayed nearly constant, decreasing from 26 to $23 \times 10^{11}$ m$^{-1}$ with age (Spinhirne et al., 1998). The derived $N_{ice}$ would be >3 times larger for C <0.7. The method would be less suitable for thick cirrus with far lower C values.

### 3.2.7 Spiral: A spiral contrail spreading in clear air

An unusual example of a spiral-contrail forming from a circling military aircraft was observed in NOAA-14 AVHRR data
west of Denmark 14:45 UTC 22 May 1998. A B707 aircraft, with four CFM 56 engines, was cruising at a speed of 660 to 720 km h$^{-1}$, at flight level FL290 (8.84 km). Schleswig radiosonde data at 12 UTC 22 May 1998 indicate that the contrail formed above 9 km altitude (p < 300 hPa, T < -47°C) at levels with moderate stratification ($N_{BV} = 0.011$ s$^{-1}$, $Sh_T = 0.001$ s$^{-1}$), where the mean wind blew at more than 110 km h$^{-1}$ from 330°. The contrail was 1500 km along nine circles of 60-km diameter along the spiral. This is the largest length identified in this study. W varied between 5 and 10 km. From the aircraft
speed and from the spiral shift by 195 km with the 110 km h$^{-1}$ wind, one estimates an age of 2 (1.7-3) h. The mean/maximum contrail $\tau$ increases linearly from 0.1/0.5 to 0.3/0.9 with age, while $\tau$ in the background air along the spiral increased up to 0.1 (Schumann, 2002; Kästner, 2003).

### 3.2.8 Cluster: Contrails merging into clusters

Duda et al. (2004) observed the merging of several contrails into a contrail cluster over the Great Lakes (Wisconsin and
Michigan) using a combination of geostationary (GOES-8) satellite, NWP, and traffic data for 14 to 19 UTC 9 October 2000. They identified individual contrails in the upper troposphere (FL 350 to 390, p = 240 to 200 hPa), at high humidity. The $\tau$ of the contrails (and contrail clusters) was about 0.25 (0.14 to 0.55). Normal ambient shear was low, at (1.5 to 2.8)$\times 10^{-3}$ s$^{-1}$. Nevertheless, high lateral contrail spreading rates were found. The trails were about 6 km wide at 2.25 h age and 10 km wide at 3.75 h age, i.e., the mean spreading rate was 0.75 m s$^{-1}$. The growth rates were explained for given shear by contrail depth
increase with time from ice particle sedimentation, and the mean fall speeds of the ice particles was estimated from this connection. Still, the derived rate is 3 times smaller than that found by Freudenthaler et al. (1995) for younger contrails. Perhaps the shear was below average in the observations of Duda et al. (2004). Also, the estimated width-spreading rates may be affected by the relatively coarse resolution of GOES data. The contrail spreading results were found helpful to test global contrail modelling (Burkhardt and Kärcher, 2009).



### 3.2.9 Shutdown: Lifecycle of isolated long-lived contrails

Starting at about 16 UTC 11 September 2001, air traffic over the United States of America (USA) was halted for a period of more than 36 h, in the aftermath of terrorist attacks. Instead of the approximately 30000 flights per day (Garber et al., 2005), only a few state or military aircraft operated over the USA, forming a few isolated, wide and well observable contrails between 37°-42°N, 73°-90°W, on 12 September. Here we report results from two ad-hoc projects, "Shutdown 1" and "Shutdown 2".

In a previous study ("Shutdown 1"), Minnis et al. (2002) identified seven contrails in a sequence of multispectral images of geostationary and polar orbiting satellites, including 4-km resolution data from GOES-8 at 75°W and GOES-10 at 135°W, and 1-km data from the MODIS on the EOS Terra satellite and the AVHRR on NOAA-14, 15, and 16 after 10:14 UTC 12 September 2001. One of the contrails was followed over a time period up to 11.5 h in a sequence of satellite images. This is the oldest individual contrail identified in this study. The actual flight routes of the contrail forming aircraft are unknown. In routine mode, the GOES observation times over the Continental USA are every 15 minutes, at 02, 15, 32, 45 after the hour. The contrail might have formed prior to the first observation, in the 15-min time interval between two subsequent GOES scenes. Thus, even higher ages cannot be excluded. Contrail W, L, and $\tau$ values of the contrails were determined from the satellite images at various contrail ages (23 data points). The contrail $\tau$ is derived using the Shortwave-infrared Infrared Split-window Technique (SIST; Minnis et al. (2011)). The SIST is day-time independent since it uses only the 3.7, 11, and 12 $\mu$m brightness temperatures and solves for cloud optical depth, contrail temperature, and particle size. Longwave radiative forcing is computed as in Palikonda et al. (2005). The results show a mean $\tau \approx 0.23 \pm 0.14$. Most of the trails spread at 7-8 km h$^{-1}$ with maximum W after 2-4 h. The mean W is $(14.5 \pm 9)$ km, and the mean L is $(205 \pm 121)$ km. For a subset of the contrails, with nearly coincidental GOES-8 and NOAA-14 and 15 observations near 11:09 and 12:45 UTC 12 September, the geometric contrail altitudes were estimated from stereography. Many of the contrails had altitudes between 9 and 12 km above mean sea level, with extreme values of 6.5 and 13 km.

As a supplement to that paper, the data set has been expanded in a new study ("Shutdown 2") by analyzing additional AVHRR and MODIS observations between 00:11 and 16:00 UTC the same day using the methods described in Bedka et al. (2013). In addition to contrail W, L, $\tau$, and altitude, the effective radius, $r_{eff}$, are also derived for 15 contrails, of which 7 overlap the observations in the Shutdown 1. One or more of the earlier contrails was likely produced by the US Air Force One aircraft (a B747 variant) and its fighter escorts, but details are unknown. Contrail temperature was derived and converted to altitude using NWP data. Contrail ages were estimated by visually searching for the first appearance of contrail traces in sequential 1-km visible-channel GOES scenes. GOES data between 23:45 UTC 11 September and 01:02 UTC 12 September and between 03:45 and 06:45 UTC were missing, while others were incomplete. Because of the low resolution of the thermal infrared GOES imagery available, RGB data were used (i.e., a combination of visible (0.65 $\mu$m), 3.9 - 11 $\mu$m, and 11 $\mu$m data during the day, and 3.9 $\mu$m, 11 $\mu$m, and 3.9 - 11 $\mu$m data at night). The detectability of the contrails in the RGB data may be slightly later in the contrail's lifecycle than for 11 - 12 $\mu$m imagery. This is especially true for contrails





that are embedded in or extend from cirrus decks, and therefore, some of the contrail ages given in this study may be underestimated.

The mean results are plotted versus contrail age in Figure 2. The mean values with error bars depict the age, i.e., the difference between the actual observation time and the time of first observability, with 0.5 h added uncertainty for the

Shutdown-1 and 1 h for Shutdown-2 results. The values plotted for $\tau$ and $r_{eff}$ are the median values with 10 to 90% percentiles. The uncertainty of W and L are estimated as ±30 and 20%. The total extinction (the integral of the optical depth over contrail width) is estimated from the product of W and $\tau$ based on the respective median values and uncertainty ranges from $\tau$ and W. The results show reasonable variations. In particular, particle sizes seem to grow systematically with age. One (Shutdown 1) contrail result stands out, likely because of underestimated contrail age.

For those contrails which were treated in both analyses, the Shutdown-2 results for $\tau$, W, altitude and L differ by about 20 to 50% from the Shutdown-1 results. These deviations are essentially within the error bars, but the Shutdown-1 results tend to have larger $\tau$ and W, while the L values are similar and correlated within 90 km for most cases. The differences between the two analyses demonstrate the general uncertainties involved in satellite retrieval of contrail $\tau$ and $r_{eff}$ in addition to technical differences. Several factors contribute to these differences. The two methods identified contrail pixels

differently. For the MODIS retrievals, the mask C from the automated contrail detection algorithm is used (Bedka et al., 2013), while the AVHRR retrievals were done with the split-window technique for 11 - 12 μm brightness-temperature difference (BTD) thresholds (Mannstein et al., 1999), subjectively throwing out non-contrail pixels by hand. The contrail pixels in most of the images were determined subjectively (except for the new MODIS analyses at 4 UTC and 16 UTC where the automated mask C was used). Thus, the contrail property statistics are expected to vary. Secondly, a different

retrieval scheme was used in the two analyses (SIRS or BTD). Thirdly, the contrail height was estimated differently in the two studies. Although the contrail heights from the later analyses are generally consistent with the stereoscopic determinations from the first analysis (with the potential exception of the MODIS results), even minor differences will lead to added uncertainty in retrieved contrail properties.

We also compared the Shutdown-2 contrail results to coincident NWP data from ERA interim (Dee et al., 2011). The

validity of the NWP data for this case was checked by comparison to two weather-balloon soundings. The temperature profiles exhibit a double tropopause for that day, one near 12 km and another near 17 km, with high humidity in layers of 1 to 2 km thickness partially exceeding ice saturation at 12 and 9 km altitude. Wind speed reached 20 m s$^{-1}$ blowing from the west between 12 and 14 km attitude. In the NWP data, temperature is rather homogeneous horizontally and slowly decreased with time during that day. Vertical profiles of temperature, wind, geopotential altitude, and even relative humidity over ice in

the mid troposphere from NWP data agree quite well with the radiosonde data. Although RHi from radiosondes and NWP may have limited accuracy at low temperatures, some altitudes (>13 km) derived from the contrail analysis appear as possibly too high because the air masses were dry (RHi below 30%) at those altitudes in the NWP analysis.





The NWP data were also used to check for source identity of contrails at various ages. Air parcel trajectories were computed using linearly interpolated winds and a second-order numerical integration scheme with about 20-min time steps, which is fully sufficient for NWP data having 3-h resolution. Starting with an observed contrail at a known position (longitude, latitude and altitude) at one time, the expected position of the same contrail at a later time is taken from this

trajectory analysis. By comparing computed and observed positions at the later time, it can be determined whether two contrails observed at two subsequent times have identical sources or not. With this analysis we confirmed that some of the contrails originated from the same source, while others, in apparently close proximity with similar morphology, had, in fact, different sources.

Atmospheric parameters ($p$, $T$, RHi, $Sh_T$, $N_{BV}$) at contrail positions were derived from the NWP data. The atmosphere

was stably stratified ($N_{BV} = 0.017$ s$^{-1}$ on average), typical for the upper troposphere. Shear was rather low ($Sh_T \approx 0.0012$ m s$^{-1}$), and can explain the large observed W only for large D, again suggesting possible enhancement by sedimentation.

### 3.2.10 Fallstreaks: Fallstreak formation for a sequence of contrails

Atlas et al. (2006) observed individual contrails for ages up to 2 h. The first set of contrails was observed between 15:30 and 18 UTC 7 September 2003 with a camera from the ground, with a lidar at Goddard Space Flight Center (39°N, 76.9°W), and

in MODIS images. The generating airliner flights were identified and used to estimate the contrail ages (1000 to 7100 s). The corresponding aircraft types are listed in the Supplement. Flight information is available from a flight track database (Garber et al., 2005), ready for a simulation of this case with proper NWP data. The observations showed contrails producing cirrus uncinus and fallstreaks. The $\tau$ was estimated as 0.35, with a maximum of 2. The total ice water content TWC per contrail length was $10^3$ to $10^4$ times the water mass emitted by a typical airliner.

The second set of data was observed between 12:00 and 22:45 UTC 5 December 2008 at the same place, again with lidar and GOES images, and analyzed using radiosonde and NWP data (Atlas and Wang, 2010). About 200 thin contrails were observed, while one contrail of about 1-h age, clearly visible with W = 10 km in GOES images, and D = 1.5 km in lidar data, reached $\tau = 2.3$. The data show the large range of variability in IWC, $\beta_{ext}$, $\tau$, and D. The NWP data allow specification of wind, T, p, Sh, and $N_{BV}$. The ice particle concentrations and the aircraft types were not determined.

### 3.2.11 SCOUT-O3: Long-lived contrails in the tropical stratosphere and above Hector

In addition to the in-situ data taken for Geophysica contrails as described in Section 3.1.11, contrail data are derived from a photo and from Falcon lidar taken during SCOUT-O3 (Schumann and co-authors, 2016). The photo was taken from the Falcon cockpit at 11.9 km altitude and shows the Geophysica contrail at 17 to 19 km altitude in the stratosphere at 8:35 UTC 16 November 2005 west of the Tiwi Islands. The position of the Geophysica exhaust plume responsible for the observed

contrail could be reproduced from the Geophysica flight path and wind data. The Geophysica was descending along a flight path with several curves from about 19 km in the stratosphere to 17 km altitude at the tropopause while it formed a visible





contrail in clear air. Two contrail-segments, above and below 18 km altitude, were distinguished in the photo. The optically thin persistent contrail segments were 1200 to 3200 s old, 1-3 km wide, each about 200 km long, and originated in ambient air with mean T and RHi values and standard deviations along the flight paths of T = (-82±0.6)°C and (-85±2)°C, and RHi = (64±7)% and (87±23)%, respectively.

Lidar data were obtained with an upward-looking backscatter lidar on the DLR Falcon for the "Golden Hector" day of 30 November 2005. The lidar observed two cirrus layers from the anvil of a deep convective cloud, containing Geophysica exhaust, in particular in the upper of these two cirrus layers. The anvil is also seen in two NOAA AVHRR scenes, but line-shaped contrail structures were not identifiable. The geometrical and optical thickness of the cirrus layers and their geometrical width as well as the total extinction (integral of optical depth over width along the Falcon flight path) was

derived. The atmospheric conditions for these cirrus layers were taken from the in-situ measurements on the Geophysica at the time of contrail generation. The age of the contrails was derived from trajectory analysis. The observations represent mixtures of natural cirrus with contrails.

### 3.2.12 ACTA: Lifecycle and radiative properties of satellite-observed contrails

Vázquez-Navarro et al. (2010) developed the Automatic Contrail Detection Algorithm (ACTA) to identify contrails in

simultaneous MODIS and METEOSAT satellite scenes and to subsequently trace the contrail history forward and backward in time in rapid scan (5 min) METEOSAT images with 3 km × 3 km spatial resolution at nadir. ACTA was developed within DLR projects (CATS and WeCARE). A total of 2400 contrails was observed with the ACTA method, at multiple times during their lifecycles, mainly over the Atlantic and Europe at northern midlatitudes, from August 2008 to July 2009, yielding more than 25000 data entries for contrail L, W, $\tau$, height, shortwave and longwave local radiative forcing and

energy forcing (Vázquez-Navarro et al., 2015). The given ages do not include any dwell time needed to make the contrails visible in the coarse Meteosat scenes (about 0.5 to 1 h). Mean (maximum) values retrieved are 1(17) h duration, 130 (>400) km length, 8 (14) km width, 11.7(14) km altitude, and 0.34 (2) optical thickness. A table with the ACTA results and the percentiles of the $\tau$ data is included in the Supplement.

Figure 2 includes a graph of the median values and the $\tau$ percentiles. The optical depths in the ACTA data are within the

upper range of other observations, as to be expected (Schumann et al., 2015). The derived width values are not plotted; they are large also for young ages, and seem to suffer from the coarse satellite pixel resolution. The data would allow extraction of the relevant meteorological parameters from NWP data. The individual aircraft causing the contrail have not been identified, because air traffic data are missing. The local RF values are of order ±20 W m$^{-2}$, consistent with other observations and model results (Duda et al., 2001; Haywood et al., 2009; Schumann et al., 2015).





### 3.2.13 Cameras: Contrails observed with several ground-based cameras

Four individual persistent contrails behind well-identified aircraft (small and large ones) were tracked in clear air with several ground-based video cameras 23 to 37 km apart (Schumann et al., 2013a), as a contribution to the DLR project WeCARE. Contrail properties in terms of altitude and visible width W were derived up to a maximum age of 2700 s. The data fit well with the other W results (Figure 2). The ambient meteorology was characterized with radiosonde and NWP data. The contrails were compared with individual CoCiP contrail simulations, and it was found that the W (e.g., 1500 m at 1400 s age) was roughly simulated as observed. One of the contrails formed during aircraft climb in clear air at a temperature, derived from ECMWF data, of about -39°C, more than 7 K above the SAC temperature.

### 4 Discussion

### 4.1 Contrail properties versus age

Figure 3 depicts the data from all observations collected in COLI for various local contrail properties together with results from CoCiP as a function of contrail age. Nearly 90% of the contrail cases occurred at temperatures between -60° and -40°C, but the data set includes contrails at temperatures as cold as -88°C (CR-AVE) to as warm as -31 °C (ICE-1989), and at altitudes between 7.5 km and 18.7 km. The observed jet aircraft are of different sizes, with 12 to 79.8 m wing span, 2.5 to 508 Mg mass, and 0.57 to 16 g m$^{-1}$ fuel consumption per flight distance.

When discussing trends in these data with age, one must note that the plots show the properties of the ensemble of contrail observations versus ages at the times of observations, i.e., in a quasi-Eulerian mode. Different trends may occur when observing single individual contrails during their evolution with time, in a quasi-Lagrangian mode. Only a few observations provide information on individual contrails in a Lagrangian mode. The CoCiP model results used here are also for an ensemble of contrails observed in the Eulerian mode. The ACTA data include data for individual contrails at a sequence of times and data for the ensemble of contrail observations.

Both the observations and the model results show a large range of variability in the contrail properties that increases with age. The results from the measurements and simulations agree on the mean magnitude. The observations may be biased somewhat to thicker contrails, because they are typically easier to observe. The trends are as expected (Schumann and Heymsfield, 2016). Ice-particle number concentration $n_{ice}$ is high in the young exhaust and then decreases (with power -1.6 of age in these data), mainly because of dilution. The remote sensing results of Spinhirne et al. (1998) and Duda et al. (1998), blue points in Figure 3a, are within the spread of the in-situ data. The IWC (Figure 3b) adjusts to the available supersaturation, which is temperature dependent and appears to decrease slightly with age. The particle sizes (Figure 3c) are initially small, but grow with age, depending on the number of particles competing for the available humidity. The effective radius $r_{eff}$ is often larger than the volume mean radius $r_{vol}$, and $r_{eff}$ grows slightly faster with age (with power 0.5) than $r_{vol}$ (power 0.33), presumably because the size distributions broaden with time with larger separation between sizes of maximum





volume and maximum number densities for aged contrails. The ratio $C = r_{vol}/r_{eff}$ ranges from 0.8±0.2 for young contrails to 0.6±0.4 for about 1-h aged contrails. The optical depth (Figure 3d) is largest for young contrails and then approaches a near-constant median value of about 0.3. The contrail width W (Figure 3e), reaching up to 40 km, is controlled by the aircraft wake in the first ~100 s, and then grows almost linearly over two orders of magnitude, mainly because of horizontal wind

shear. The contrail depth in Figure 3f is the maximum depth D derived from the observations. It is larger than the effective depth $D_{eff} = A_c/W$, controlling $\tau$ or ice water path. D is controlled by vertical mixing in the wake vortex initially and then grows less strongly (power 0.3, over one order of magnitude) mainly by turbulence and particle sedimentation when the particles reach a sufficiently large size. The D/W ratio values vary over a wide range similar to those derived from a combination of satellite and lidar data from MODIS and Cloud-Aerosol Lidar and Infrared Pathfinder Satellite Observations

(CALIPSO) (Iwabuchi et al., 2012).

Figure 4 shows integral contrail properties (and length and $\beta_{ext}$) versus age. Integral properties have often been computed from models (Unterstrasser and Gierens, 2010a; Lewellen et al., 2014). This is the most complete set of integral contrail data derived from observations to date. The figure includes results from remote sensing and a few in-situ measurements with estimated cross-sectional areas. The contrail cross-sectional area $A_c$ in Figure 4a grows with a power of

1.4 of age by turbulent mixing, enhanced by shear, ice particle sedimentation, and radiative heating. It could diminish when part of the contrail evaporates, but this effect is only weakly indicated in this collection (e.g., at large ages). The total number of ice particles per contrail length (Figure 4b) is large initially because it is mainly controlled by exhaust particles and decreases slowly with age. This confirms earlier suggestions (Spinhirne et al., 1998) and is remarkable, because it suggests that any ice nucleation process contributes little to contrail ice particles at later ages. This does not exclude re-nucleation of

ice particles from the aerosol released from the contrail after sublimation (Schumann and Heymsfield, 2016).

The particle losses at later ages certainly include ice sublimation in dry air and precipitation. More complex processes like aggregation (Kienast-Sjögren et al., 2013) or humidity fluctuations under the influence of the Kelvin effect (Lewellen, 2012), may be present but are not obvious in these observations. The total ice water content TWC (Figure 4c) varies strongly with large variability over about 6 orders of magnitude. Low TWC values may result when the ambient air dries (e.g., in

subsiding air masses). Large TWC values develop in rising air masses with increasing relative humidity, for strong mixing with humid air masses, and for sufficiently large ice particles falling into more humid layers. Sedimentation contributes to a dehydration of the upper troposphere (Burkhardt and Kärcher, 2011; Schumann et al., 2015).

The total extinction EA (Figure 4d) multiplied by lifetime or length controls the contribution of an individual contrail to warming or cooling the Earth atmosphere. EA grows about linearly with age. Hence, when multiplied with lifetime or length,

the product (related to the product of $\tau$ and contrail coverage) varies over about 7 orders of magnitude. The amount of energy put into the atmosphere by a contrail (the energy forcing (Schumann et al., 2012)) is proportional to EA but depends further on radiative boundary conditions and hence varies in sign, and its magnitude varies over an even larger range than the local contrail properties.





The contrail length (Figure 4e) reaches 620 km for the linear contrails and 1500 km for the spiral contrail in this collection. The length is of course limited by aircraft speed and age, but also by the meteorological variability along the flight path. Length values have rarely been derived from in-situ observations. Remote sensing studies (Iwabuchi et al., 2012; Vázquez-Navarro et al., 2015) provide more extensive data bases for length, but with similar maximum value of 1200 km.

The $\beta_{ext}$ values included in this plot show a quite consistent picture, with values, in general, decreasing slowly with age. The $\beta_{ext}$ values (Figure 4f) should increase for growing ice particles and decrease for decreasing ice particle concentrations. In addition, they depend on the particle habit, which may be quasi spherical in young contrails (at least at low temperatures (Freudenthaler et al., 1996)) and become more complex in aged contrails. Apparently, the dilution effect dominates the $\beta_{ext}$ change with age.

Some of the mean contrail properties are specifically sensitive to one set of observations only. In particular, this is the case for remote sensing of $\tau$ and of $r_{eff}$, which are determined mainly by the IFU-Lidar data for ages < 1 h and by the Shutdown data for larger ages.

### 4.2 Humidity and contrail ice water content versus temperature and contrail threshold

Figure 5 (upper panel) shows that the IWC in contrails scatters over a wide range of values and varies strongly with
temperature, as expected. The linear regression for the in-situ data is close to an early parameterization by Schumann (2002). Relative to the regression line, the logarithmic IWC values vary with a standard deviation which corresponds to a factor 3.8 for linear IWC values. The thin black curves represent maximum, median, and minimum IWC values from a large set of in-situ measurements in cirrus (Schiller et al., 2008). Extreme values of contrail IWC have been observed rarely, with minimum/maximum values reasonably within the range of IWC measured for other cirrus. The results from SCOUT-O3 and
CR-AVE nicely extend the previous results for CRYSTAL-FACE to lower temperatures, without any notable outliers. In fact, the extreme values are quite understandable. The three low IWC values between 218 and 221 K come from CONCERT 2011, possibly with sublimating ice, including a thin A321 contrail at the end of the wake vortex phase at 91 to 95% RHi (Kaufmann et al., 2014). The high in-situ value at 235 K is from the PMS results (Knollenberg, 1972) for a contrail resulting in cirrus uncinus. Similarly, the high remote-sensing value at 215 K apply by for a thick fallstreak contrail (Atlas and Wang,
2010). The high in-situ values between 185 and 192 K are from the ice events measured during SCOUT-O3 (de Reus et al., 2009), which might be contrail cirrus. The ice events contain many small ice particles, possibly from the Geophysica aircraft exhaust, and a few large ice particles, that dominate the IWC, due to deep overturning convection. The two sets of IWC values for CRYSTAL-FACE contrails, larger from Mullins (2006) and smaller from Gao et al. (2006), are both within the range of other data. In some studies (Knollenberg, 1972), the IWC is higher than to be expected from deposition of ambient
ice supersaturation if the ambient air was saturated with respect to liquid water. This may come from non-adiabatic effects, like particle growth during sedimentation, uplifting motions, or mixing with ambient cirrus.

The lower panel of Figure 5 shows the ice-equivalent relative humidity over ice,





$$RHx = (M_{air}/M_{H2O}) \, (IWC/\rho)/(p_{ice}(T)/p), \qquad (4)$$

for given molar masses of air and water, ice water mass per unit air mass, air density, ice saturation pressure for given temperature and pressure. We see, RHx is larger at low temperatures than at high temperatures. This is to be expected (Schumann, 2002), because the humidity difference between liquid saturation and ice saturation decreases with increasing temperature and is zero near 0°C. The RHx difference between homogeneous ice nucleation and ice saturation is smaller and zero above -38°C (Koop et al., 2000). In a few cases, the observed RHx values are very large. These are cases with strong diabatic ice formation, as discussed before.

Figure 7 shows histograms of RHi in the ambient air for all the data (including remote sensing data, with RHi from radiosondes or NWP data) and for the in-situ data only. We see maximum values of 165% and many cases near ice saturation, but also many cases where the available information suggests ice subsaturation. Certainly, the accuracy of the given RHi data varies. Contrails (and cirrus) are observed frequently in apparently subsaturated air (Jensen et al., 2001; Ovarlez et al., 2002; Krämer et al., 2009; Kübbeler et al., 2011). The common explanation is that these are clouds with large ice particles that have been formed under supersaturated conditions, have sedimented to lower and drier levels, and are sublimating slowly. A process-oriented study of measured contrail ice in subsaturated air is missing. Contrail model simulations often depend strongly on accurate ambient humidity data (Sussmann and Gierens, 2001). If such data are missing, one may consider adjusting ambient humidity such that the simulated contrails reach the observed ice water content.

The mean IWC and RHx trends are most sensitive to the observations at the highest (ICE-1989) and lowest (CR-AVE and SCOUT-O3) temperatures. If these data are excluded from Figure 7, the mean trends in both panels stay about unchanged but the confidence ranges double.

## 4.3 Consistency with the Schmidt-Appleman Criterion

Figure 6 shows the temperature difference between ambient temperature T and the SAC threshold temperature $T_{LC}$. (Schumann, 1996; Jensen et al., 1998b). The threshold temperature is computed for given pressure, best-estimate of propulsion efficiency (aircraft and operation dependent, between 0.1 and 0.4, mostly 0.3), combustion heat of 43.2 MJ kg$^{-1}$, and H$_2$O emission index of 1.24. The figure shows two sets of results, for observed relative humidity, and for liquid saturation; the latter are the lower values. For warm air with T > -40°C, we see that 15 of the contrails were observed at ambient conditions above the SAC threshold. Assuming liquid saturation, this number reduces to 7. The differences reach up to 12 K (7 K for liquid saturation), and can hardly be explained with instrument errors. Sometimes the temperature may have been measured at the contrail level while the contrail formed at higher altitude with lower temperature. Also added humidity from ambient cirrus particles entrained into the young exhaust plume from ambient air cannot explain all cases, because some of the cases were observed in clear sky (Section 3.2.13). There are a few other cases that were discussed in the literature for which contrails were observed at temperatures significantly above the SAC threshold in warm air (Gayet et al., 1996a; Mazin, 1996; Jensen et al., 1998b; Schumann et al., 2013a). Perhaps, these are aerodynamic contrails (Gierens et al.,





2009), or contrails forming on exhaust particles suitable for ice formation below liquid saturation at these temperatures (Mazin and Heymsfield, 1998), or contrails forming on pre-activated aircraft exhaust aerosol (Schumann and Heymsfield, 2016). On the other hand, there are several observations of aircraft in operation without forming a contrail, consistent with the Schmidt-Appleman criterion (and the requirement for liquid saturation in the plume), at least when one takes the effect of

propulsion efficiency correctly into account (Busen and Schumann, 1995; Mazin, 1996; Schumann, 2000; Schumann et al., 2000).

### 4.4 Measurement issues

Some of the studies found systematically higher or lower ice number concentrations. This can be seen in Figure 3a, but is more obvious in Figure 8. It shows the ice number concentrations normalized by air density to mass concentrations and

multiplied with dilution $N_{dil}$ (age) from Eq. (1), so that they can be interpreted as an apparent ice particle emission index PEI. Note that an underestimate of dilution causes an underestimate of the computed PEI. Higher dilution may in fact have occurred for some of the cases observed, e.g., for strong shear and sedimentation-driven contrail widening. In principle, one should normalize to the concentration of a fuel consumption tracer (such as $CO_2$), but such data are rare in these observations. Here, the dilution formula is approximate, and hence these values approximate the PEI.

Rather high ice number concentrations and PEI values are found for SULFUR, in particular at 120 s age for case B1 of Schröder et al. (2000). Note that the high estimated PEI value may also result from an overestimate of dilution at this age. Low values are seen for younger contrails, likely because of instrument limitations in detecting high ice particle concentrations. Low concentrations are also seen for FIRE, ICE-1989, COSIC, CR-AVE, and SCOUT-O3. The first two of these experiments in 1989 and 1991 used FSSP-100, with a lower size threshold of 2 μm, which may have undercounted

small ice particles. For young contrails, ice particles may occur even below the size range of about 0.3 μm presently feasible for modern instruments. The maximum measured concentration ($n_{ice}$ = 7000 cm$^{-3}$ at 3.5 s age) still seems small for young contrails for which concentrations as high as $10^4$ to $10^5$ cm$^{-3}$ are expected from model studies for 1-s old contrails, at least, for soot number emission indices $>10^{15}$ kg$^{-1}$ (see Fig. 2 in Kärcher and Yu (2009)). This may indicate an upper detection limit of the particle probes caused by coincidence.

Baumgardner et al. (2016) and McFarquhar et al. (2016) discuss the many issues that are associated with measuring ice particles with in-situ instrumentation. For example, ice crystal shattering on probe inlets and extended arms (Korolev et al., 2011) is a potential issue for contrail measurements. Comparisons of in-situ data with remote sensing data, however, indicate that not all cirrus with small mean effective radius can be explained by shattering (Cooper and Garrett, 2011). Shattering was also discussed for contrails (Febvre et al., 2009; Gayet et al., 2012b). E.g., Voigt et al. (2011) describe a contrail embedded

in an optically thin cirrus cloud. Here, the ambient cirrus contributed less than 1% to the number, surface, and volume distributions detected in contrails. No significant difference in the particle size range of 0.4 to 12 μm between in-cirrus and out-of-cirrus contrail observations was found. As explained in McFarquhar et al. (2016), techniques have been developed to





decrease measurement uncertainties and to minimize errors. For example, in COSIC, Jones et al. (2012) used the measured particle inter-arrival times to eliminate high particle counts (above 50 cm$^{-3}$) due to presumed shattering. They noted, however, that the high counts could be due to smaller aerosols from the aircraft plume mixed with the contrail, and noted that their low-time resolution NOx measurements did not provide enough evidence to decide if these were plume encounters.

If the non-filtered COSIC concentration data were included in our figures, they would fit better to the other observations. Shattering is not the only factor that can affect the measurements. Others include the size-sensitivity of the probes (e. g., the effect of minimum detection threshold), the sample volume uncertainties in the OAPs, particularly for sizes < 50 μm, and the coincidences of particles in the sample volume given the very large concentrations, and many more.

The PEI results (Figure 8) increased in the reported data between 1989 and 1993 (with the noteworthy exception of
CSAE). The more recent measurements support a PEI value between $10^{14}$ and $10^{15}$ kg$^{-1}$. The possible underestimate of dilution implies that the true values may well be a factor of ten higher. Nevertheless, the consistency of these data with remote sensing observations and the other measured data (e.g., $\beta_{ext}$ and particle sizes), suggests that shattering is a minor issue in these measurements. The low PEI data for CR-AVE and SCOUT-O3 are based on measurements at the lowest temperatures having very low ambient humidity. Possibly, these measurements underestimate the number of small ice
particles because they were too small. This is further supported by the fact that the highest ice particle concentration was measured during CR-AVE in the oldest part of the plume almost 1000 s after it formed. Interestingly, the PEI values for CRYSTAL-FACE, at 10 K higher temperatures, are far higher, which is not easy to explain.

By definition, the IWC should be consistent with (4/3) $\pi$ $\rho_{ice}$ $n_{ice}$ $r_{vol}^3$, within the expected uncertainties in assumed ice crystal volume and density (Baumgardner et al., 2016). This appears trivial but is not always the case. In particular some
early Falcon results (Schröder et al., 2000) deviate from this expectation by factors up to 4. Baumgardner et al. (2016) suggest that the uncertainty of IWC derived from single particle spectrometers may be as large as a factor 3, but in contrails, where there is less variation in size, shape and density, the uncertainty should be less than a factor of 2. Perhaps, experimenters have estimated IWC, $n_{ice}$, and $r_{eff}$ using different approaches to correct for known observational deficiencies, like underestimated contributions in certain particle size ranges or indications of particle shattering. Some authors may have
used other radius definitions, e.g., with variable effective density (Heymsfield et al., 2002).

The few remote sensing data for ice number concentration fit reasonably to the in-situ data. In particular, the contrail properties derived from the Shutdown-2 analysis are consistent with the other observations. If one derives the number of ice particles per contrail length from $N_{ice}$ = EA/($Q_{ext}$ $\pi$ $r_{area}^2$) for these data, the $N_{ice}$ vary in the range of (4 to 50)×$10^{11}$ m$^{-1}$, depending on the assumed ratio $r_{area}/r_{eff}$ = C$^{3/2}$, with C of about 0.5 to 0.7. This range of scatter is small in view of the fact
that very different aircraft sizes and types caused the contrails. The result is not much different from that of Spinhirne et al. (1998). Hence, these studies provide further evidence for a nearly constant number of ice particles in contrails.

Several of the contrail properties show systematic changes of mean values or trends with age or temperature, such as EA with age and IWC with temperature. The mean values and trends are sensitive to data from a few observation projects,





such as the IFU-Lidar and Shutdown satellite analyses for the change of $r_{eff}$ and EA with age, and ICE-1989, CRYSTAL-FACE, CR-AVE and SCOUT-O3 for the change of IWC with T. However, none of these results changes significantly when one data source is excluded. For SCOUT-O3, we repeat that these data are from potential contrail self-encounters.

### 4.5 Model comparisons and open issues

The observations generally agree well with the model results. Both show a large range of variability and changes with age and agree on the median magnitude. The observations are all within the range of model results, at least if we include the minimum and maximum values of the CoCiP results. CoCiP also reflects properties of thinner contrails, less easy to observe. Figures 1-4 compare contrails for very different aircraft and atmospheric conditions. A similar comparison, for ages up to 1200 s and for a subset of the observations, shown by Naiman et al. (2011) for large-eddy simulation (LES) results, showed comparable, but not better observation-model agreement. More demanding would be a comparison between simulated and observed results, case by case for fixed model properties. Tests for young contrails (Schumann, 2012) have shown that the agreement depends on details of ambient humidity, shear, and soot number emissions, and the remaining observations provide a weak model test when the open input is optimally adapted. Tests for young and old contrails with one set of model parameters might be more constraining. A case-by-case comparison can be done only when aircraft and atmospheric parameters are sufficiently well specified. As noted before, neither the contrail-generating aircraft nor the meteorology was identified for all cases. Most suitable for comparisons are those cases with a rich set of observations coupled with known aircraft and ambience properties. A notable example is found in the Falcon observations of the A380 case during CONCERT.

Substantial losses in the total number of contrail ice particles $N_{ice}$ have been predicted, e.g., by LES, because of adiabatic warming in the sinking wake vortex (Greene, 1986; Sussmann and Gierens, 1999; Lewellen and Lewellen, 2001; Unterstrasser et al., 2008) at contrail ages up to about 300 s (Unterstrasser, 2016). These simulations assume that ice particles that have sublimated do not leave ice nuclei which can be reactivated when the ambient humidity increases. Therefore, we discuss the observations in respect to these predicted losses, below.

Local ice concentrations $n_{ice}$ (Figure 3a) show a decrease with age both before and after the end of the wake vortex phase, but this does not reflect changes in the total number. There are only a few observations of total ice number concentrations $N_{ice}$ available (Figure 4b). More data are available for EA (Figure 4d). The EA data show a gap between ages of 100 to 300 s, which is interesting. A comparable plot from LES results, Fig. 2c of (Lewellen et al., 2014), shows a similar but weaker minimum of EA (actually $EA/Q_{ext}$ is plotted) near 100 s age for the LES case with maximum particle losses.

As explained in Jeßberger et al. (2013), EA scales as

$$EA = [9\pi/(16\,\rho_{ice}^2)]^{1/3}\,C\,Q_{ext}\,m_F\,[f_s\,PEI_{ice}]^{1/3}[EI_{H2O} + (RHi\text{-}1)N_{dil}\,M_{H2O}\,p_s(T)/(M_{air}\,p)]^{2/3}. \qquad (5)$$



Most symbols were explained before: $\rho_{ice}$ = ice bulk density; C = $r_{vol}/r_{eff}$, $Q_{ext}$ = extinction efficiency, $m_F$ = fuel consumption per distance, $f_s$ = survival factor of ice particles, $PEI_{ice}$ = ice particle emission index, $EI_{H2O}$ = water vapor emission index, RHi = relative humidity over ice, $N_{dil}$ = dilution factor, $M_{H2O}$ and $M_{air}$ = molar masses of water and air, $p_s(T)$ = saturation pressure, p = air pressure. $Q_{ext}$ may increase with age when the initially sub-micron sized ice particles grow much larger than the wavelength of light, $N_{dil}$ increases strongly with age, and $f_s$ should decrease for strong particle losses. So any decrease of EA with age during the wake vortex phase could be an indicator for particle losses.

The dark green curve in Figure 9 shows an evaluation of the above total extinction model versus age for T = 220 K, p = 240 hPa, C = 0.7, $m_F$ = 5 g m$^{-1}$, $PEI_{ice}$ = $10^{15}$ kg$^{-1}$, for constant survival factor $f_s$ =0.7, and a new contrail dilution model $N_{dil}$ = 20 $(t/t_0)^{1.7}$, $t_0$ = 1 s, with coefficients (in particular exponent 1.7) fitted to the observed growth in cross-section area. The constant, $f_s$, accounts for mean losses, but does not describe any temporal evolution of particle losses. This EA model fits the remote sensing observations (mainly from IFU Lidar for ages < 1 h and from Shutdown data for larger ages) reasonably well for constant $f_s$, but without the gap or local minimum which seems to exist in the observations near 100 to 300 s. Hence the observations may be sensitive to particle losses in the wake phase; however, a final conclusion is still not possible. A closer investigation should check for any other reason that may explain the gap (e.g., high EA in young contrails because of strong vertical mixing or high temperature, and high EA at large ages because of re-nucleation of residuals from sublimated ice particles in supersaturated air).

## 5 Conclusions

A "contrail library" COLI has been set up, collecting microphysical and geometric contrail properties from past measurements for individual jet aircraft contrails with known ages, together with information on the aircraft and atmosphere when available. The data base includes more than 230 entries describing properties of contrails with known ages, including mean data for 100 cases of in-situ measurements, more than 70 cases from ground-based and airborne lidar observations, 50 cases from satellites, and a few camera observations. The data come from 33 observation projects during the last 45 years.

The comparison of the data from various measurements shows notable differences among the various measurements from different instrumentations. But generally, the agreement among each other and between the in-situ and remote sensing results, and also the agreement with the model results, shows that the basic properties of contrails are quite well documented. The comparisons provide some answers to the questions listed in the Introduction. For example, all results are consistent with the assumption that contrail ice particles form from exhaust in the young contrail and the number does not change much later during the lifetime. The data show the high importance of cirrus uncinus formation with fallstreaks in ice-supersaturated environments. These cases are responsible for large D, W, $r_{eff}$, and IWC. The comparison gives strong support for the general validity of the CoCiP model for the period after the wake vortex phase. The apparent emission index of ice particles may exceed $10^{15}$ kg$^{-1}$, because the computed PEI values are dilution dependent, and dilution in contrails was stronger than measured in exhaust plumes without contrails. The ice water content in contrails, on average, follows a previously suggested





negative correlation with temperature and higher ice-equivalent relative humidity for low temperatures, as expected thermodynamically and from homogeneous ice nucleation theory. The ratio of volume to effective mean radius decreases with time, which affects the optical depth and the radiative effectiveness of cirrus computed for given number density and ice water content. The optical extinction decreases with contrail age, mainly because of particle dilution. Surprisingly,

several contrails were observed in warm air above the Schmidt-Appleman threshold, without full explanation.

There is no obvious indication that ice crystal shattering on instruments has significant effects on past contrail measurements. Shattering may still be important in warm contrails with a high number density of large ice crystals. In nascent contrails and at low temperatures, the measurements may have underestimated the concentration of sub-micrometer ice particles. The remote sensing contrail data, though also not without uncertainties, support the general consistency of the

data obtained. More data, from various methods and regions, would allow reducing the sensitivity of mean results to single observation projects.

In view of the large variability, the total amount of data is still small. More data and more detailed comparisons of model results with the observations are needed for final answers to, e.g., the questions raised. Models should be run for case studies of ensembles of contrails in an Eulerian mode at various ages and for individual contrails in a Lagrangian mode as a

function of age, and measurements should be interpreted for these modes separately, to see whether the derived correlations describe the full life cycle of contrails. A future study similarly could collect and compare contrail-cirrus data.

In many cases, the relative humidity inside and outside the contrails was low and not consistent with the measured ice water content, and the data missed accurate quantification of dilution. This data base should be supplemented with future measurements providing "complete" data sets for individual contrails, with data on all the parameters listed in Tables 1 to 6.

This includes data on contrail age and geometry, source aircraft, microphysics, tracers of dilution, ambient atmosphere parameters (temperature, humidity, shear, stratification, turbulence, radiation, ambient clouds), and the instrumentation. In addition, there remains the need for new instruments that can successfully measure the high concentration of small contrail crystals near the source, overcoming the current limitations of particle detection at wavelengths of visible light and coincidence in the sample volume.

*Acknowledgments.* We are grateful to input and comments from William A. Cooper, Ru-Shan Gao, Timothy Garrett, Kaspar Graf, Christian Kiemle, Andreas Minikin, Andrew Roberts, Daniel Sauer, and Simon Unterstraßer. S. Bedka, D. Duda, and P. Minnis were supported by the FAA ACCRI Program. We gratefully acknowledge the wide set of data sources as explained in the text. The first author dedicates this work to his former colleague, Dr. Manfred E. Reinhardt (26 January 1927 to 1 October 2015), a reliable partner, for fruitful cooperation 1982-1991, and for early contrail research with research

aircraft.





**Appendix A: Early measurements on the formation conditions of contrails**

Early flight experiments on the formation condition of contrails were performed in several countries (Brewer, 1946; Appleman, 1953; Mazin, 1996; Schumann, 1996). First in-situ measurements in the exhaust plume of a contrail forming aircraft were performed in Germany since 1939 (aufm Kampe, 1942, 1943). The aircraft was a single-propeller Henschel

(Hs-126; span 14.5 m, mass between 2 and 3.2 Mg), with a Bramo-Fafnir motor, flying with about 135 kg h$^{-1}$ fuel consumption ("Oktan 87"), at 8000 to 10000 m altitude, at 66 m s$^{-1}$ speed. About 60% of the combustion energy was expected to heat the engine exhaust, so that the overall propulsion efficiency is about 0.4. The water content of the fuel was higher than for kerosene, but the fuel is not burned completely, so that the effective water mass emission index was estimated to be ~1.25. Temperature above ambient air and air speed were measured in-situ on a towed glider aircraft and at

the wing. The plume excess temperature was maximal (36 K) about 1.5 m behind the engine exhaust exit, about 7 m before the aircraft tail, 14 K at the tail, and about 3 K at 60 m behind the tail. The measured propeller wake speed was 77 m s$^{-1}$ (about 11 m s$^{-1}$ higher than the aircraft speed) at a position 6 m behind the engine and 69 m s$^{-1}$ at 60 m distance. The humidity inside the exhaust plume was computed for the given fuel consumption, emission index, and measured wake diameter. The temperature was measured, and it was found that liquid saturation was reached in the propeller wake, about 10

to 20 m behind the tail. Contrail formation was observed and documented in photos from escorting aircraft and from ground. The distance increased with ambient temperature, consistent with the thermodynamic theory.

Frost was observed on the towed glider, indicting some liquid water in the young contrail during these experiments. The dilution in the propeller wake is about 10 times larger than expected for jet airliners from Eq. (1). We include one data line from these reports, listing the diameter of the contrail as ~3 m at contrail formation, 20 m or 0.26 s behind the aircraft tail,

but ambient conditions for these observations have to be estimated (about -45°C ambient air temperature at 8 km altitude). The data are not included in the plots because they are from a propeller-driven aircraft, which may show stronger dilution than a jet aircraft.

From the observations of contrail persistence during 150 flights, it was concluded that ice supersaturation occurred frequently while water supersaturation was assessed to be unlikely (aufm Kampe, 1943). The size and shape of contrail ice

particles was derived from impactor collections in the contrails and contrail cirrus; see also Weickmann (1945). Some of the contrail particles grew large and formed fallstreaks. Halos were observed in the contrails indicating unimodal hexagonal ice crystals, bullets, or plates (aufm Kampe, 1942, 1943; Weickmann, 1945).





**Appendix B: On the Contrail Cirrus Prediction Model CoCiP**

The plots include results from CoCiP as taken from the simulations described in Schumann et al. (2015). The method is documented in Schumann (2012). This appendix briefly summarizes the model and simulation parameters relevant for the comparison to observations.

CoCiP is a Lagrangian model that simulates contrail segments for each flight segment of a given aircraft flight trajectory and for the state of ambient atmosphere (temperature, humidity, wind etc.) interpolated to the flight track waypoints in time and space. A contrail is initialized when the Schmidt-Appleman criterion is satisfied. The contrail life ends in the model when the contrail particles precipitate below the lower boundary of the computational domain (at about 450 hPa) or when the $\tau$ becomes smaller than a low threshold ($10^{-6}$). Hence most of the contrails analyzed formed in ice-

supersaturated air, but contrails occur also in subsaturated air (either short-lived contrails or older contrails with large cross-section and still finite ice water content). The initial number of contrail ice particles corresponds to a modelled fraction (~0.5 to 1, depending on aircraft, emissions and ambient conditions) of the soot particles emitted. The simulation was performed for an assumed constant soot particle number emission index of $10^{15}$ kg$^{-1}$, assuming that each soot particle nucleates one ice particle. The atmospheric parameters are taken as provided by the general circulation climate model CAM-3 with ice-

supersaturation microphysics (Yun et al., 2013), except that humidity was increased by a factor 1/0.9. The results shown were obtained in the coupled model version that includes exchange of humidity between the background atmosphere and contrails in both directions. The weather considered in the model is the weather for the last 30 years simulated in the climate model. Each individual contrail is computed for a given aircraft type and aircraft properties (flight level, position, span, mass, fuel consumption, speed, propulsion efficiency) as derived from the ACCRI traffic waypoint data set for the year 2006

(Brasseur et al., 2016), using BADA data to estimate aircraft mass and fuel consumption (EUROCONTROL, 2009). A total of $3\times10^7$ contrail segments are simulated per year and analyzed hourly. The data were used to compute median values and percentiles for plotting. A table with the CoCiP percentiles as shown in Figures 1-4 is included in the Supplement.

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



**Tables**

**Table 1.** Aircraft and Engine

| Symbol | Property | Unit |
|---|---|---|
| ATYP | aircraft type | text |
| Engine | engine type | text |
| $N_{engine}$ | number of engines | 1 |
| span | wing span | m |
| z | altitude above mean sea level (MSL) | m |
| FL | pressure altitude, flight level in hft, 1 hft = 30.48 m | hft |
| p | pressure at flight level according to ICAO standard atmosphere | hPa |
| M | aircraft mass | Mg |
| TAS | true air speed | m s$^{-1}$ |
| heading | true heading (flight direction), 0° = northbound, 90° = eastbound | degree |
| $m_F$ | all-engine fuel consumption per flight distance | kg m$^{-1}$ |
| η | overall propulsion efficiency, $\eta = F/(m_F Q_C)$, F = thrust, $Q_C$ = combustion heat, typically $\eta \approx 0.3$ | 1 |
| $PEI_{soot}$ | soot number emission index: number of soot particles suitable as contrail condensation nuclei per mass of fuel burnt. Here, $10^{15}$ kg$^{-1}$ in the CoCiP simulation | kg$^{-1}$ |
| traffic data | information on source for traffic data | text |

**Table 2.** Fuel

| Symbol | Property | Unit |
|---|---|---|
| fuel | fuel type, mostly kerosene | |
| $Q_C$ | fuel combustion heat, typically 43.2 MJ kg$^{-1}$ for kerosene | MJ kg$^{-1}$ |
| $EI_{H2O}$ | water emission index: mass of water per mass of fuel, typically 1.24 | 1 |
| FSC | fuel sulfur content (mass fraction), typically 400 µg g$^{-1}$ | µg g$^{-1}$ |




**Table 3.** Atmosphere

| Symbol | Property | Unit |
|---|---|---|
| yearmmdd | date (year, month, day) of observation | text |
| UTC_s<br>UTC_hhmmss | UTC time in s and in h, min, s, since mid-night at observation | s |
| position | longitude/latitude or region | text |
| p | pressure (computable from flight level) | hPa |
| T | temperature in K | K |
| ISSR | ice supersaturated region: 1 = yes or 0 = no | 1 |
| RHi | measured relative humidity over ice | % |
| $Sh_T$ | total wind shear | $s^{-1}$ |
| Sh | wind shear normal to flight direction | $s^{-1}$ |
| $N_{BV}$ | Brunt-Väisälä frequency | $s^{-1}$ |
| Ambient cloudiness | Presence of ambient cloud (e.g., visible cirrus): yes or no or further explanation | text |

**Table 4.** Instrumentations

| Symbol | Property | Unit |
|---|---|---|
| ITYP | instrument type | text |
| min_value<br>max_value | minimum/maximum detectable values | unit of data |

5  **Table 5.** Contrail observation cases

| Symbol | Property | Unit |
|---|---|---|
| project | acronym | text |
| Author | one or several references | text |
| platform | observation aircraft, station, satellite (possibly several) | text |
| obscase | observation case identifier | text |
| source | information on data source for observation data | text |





**Table 6.** Contrail properties

| Symbol | Property | Unit |
|---|---|---|
| age | contrail age at time of observations | s |
| $n_{ice}$ | ice particle number concentration (number of detected ice crystals per unit ambient volume) | $cm^{-3}$ |
| $A_{ice}$ | projected surface area of ice particles per air volume | $\mu m^2\ cm^{-3}$ |
| $V_{ice}$ | volume of ice particles per air volume | $\mu m^3\ cm^{-3}$ |
| IWC | ice water content, $= \rho_{ice}\ V_{ice}$: mass of ice per volume | $mg\ m^{-3}$ |
| $\rho_{ice}$ | ice bulk density, typically 917 kg m$^{-3}$ | $kg\ m^{-3}$ |
| $r_{vol}$ | volume mean ice particle radius, $= [3\ IWC/ (4\ \pi\ \rho_{ice}\ n_{ice})]^{1/3}$ | $\mu m$ |
| $r_{eff}$ | effective radius $= 3\ V_{ice}/(4\ A_{ice})$ | $\mu m$ |
| $d_{max}$ | maximum particle size | $\mu m$ |
| ext, $\beta_{ext}$ | optical extinction at visible wavelengths, cross-section mean | $km^{-1}$ |
| tau, $\tau$ | optical depth at visible wavelengths, width mean | 1 |
| D | contrail depth (maximum altitude difference between contrail parts) | m |
| $D_{eff}$ | effective vertical contrail depth | m |
| W | (visible) geometrical contrail width | m |
| $A_c$ | contrail cross-section area $= W\ D_{eff}$ | $m^2$ |
| $N_{ice}$ | total contrail ice particle number per contrail length | $m^{-1}$ |
| TWC | total contrail ice mass per contrail length | $kg\ m^{-1}$ |
| EA | total extinction (cross-section integral of extinction $\beta_{ext}$) | m |




**Table 7.** Observations included in the contrail library (Ad hoc *project names* in italic).

| Year | Project name | Source aircraft | A prime reference | Measurement | Carrier | Section |
|---|---|---|---|---|---|---|
| 1971 | *PMS* | Sabreliner | Knollenberg (1972) | in-situ | NCAR Learjet | 3.1.1 |
| 1972 | CIAP | B-52 | Hoshizaki et al. (1975) | camera | California | 3.2.1 |
| 1989 | CSAE | Learjet 35 | Baumgardner and Cooper (1994) | in-situ | NCAR Sabreliner | 3.1.2 |
| 1989 | ICE 1989 | airliners | Raschke et al. (1990) | in-situ + lidar + satellite | CAM MerlinIV + DLR Falcon + DLR DO228 | 3.1.3 + 3.2.2 |
| 1991 | FIRE/ARM | C500 | Poellot et al. (1999) | in-situ | UND Cessna Citation | 3.1.4 |
| 1991 | *ALEX-F* (SiL) | B747-200 | Baumann et al. (1993) | lidar | DLR Falcon | 3.2.3 |
| 1992 | CIRRUS'92 | airliners | Strauss et al. (1997) | in-situ | DLR Falcon | 3.1.6 |
| 1994 | CIRRUS'94 | airliners | Wendling et al. (1997) | in-situ | DLR Falcon | 3.1.6 |
| 1994 | *IFU-Lidar* | airliners | Freudenthaler et al. (1995) | lidar | Garmisch | 3.2.4 |
| 1994 | SULFUR-1 | ATTAS | Busen and Schumann (1995) | in-situ | a business jet | 3.1.5 |
| 1995 | SULFUR-2 | ATTAS | Schumann et al. (1996) | in-situ | Falcon | 3.1.5 |
| 1996 | SULFUR-4 | ATTAS, A310 | Petzold et al. (1997) | in-situ | Falcon | 3.1.5 |
| 1996 | AEROCONTRAIL | airliners | Schröder et al. (2000) | in-situ | DLR Falcon | 3.1.6 |
| 1996 | SUCCESS | DC-8 | Heymsfield et al. (1998) | in-situ | NASA DC-8 | 3.1.7 |
| 1996 | SUCCESS | DC-8 | Minnis et al. (1998) | satellite | GOES-8 | 3.2.5 |
| 1996 | FIRE, SUCCESS | B777 | Baumgardner and Gandrud (1998) | in-situ | South Kansas | 3.1.8 |
| 1996 | AEROCONTRAIL | B747-200 | Sussmann and Gierens (1999) | lidar | Augsburg | 3.2.4 |
| 1996 | SUCCESS | airliners | Spinhirne et al. (1998) | lidar + radiometer | Oklahoma | 3.2.6 |
| 1998 | *Spiral* | B707 | Schumann (2002) | satellite | North Sea | 3.2.7 |
| 2000 | *Cluster* | airliners | Duda et al. (2004) | satellite | Great Lakes | 3.2.8 |
| 2001 | *Shutdown* | B747 + fighters | Minnis et al. (2002) | satellites | North-west USA | 3.2.9 |
| 2002 | CRYSTAL-FACE | WB-57 | Gao et al. (2006) | in-situ | NASA WB-57 | 3.1.9 |
| 2003 | *Fallstreaks 2003* | airliners | Atlas and Wang (2010) | lidar | Goddard | 3.2.10 |
| 2005 | PAZI-2 | airliners | Febvre et al. (2009) | in-situ | DLR Falcon | 3.1.10 |
| 2005 | SCOUT-O3 | Geophysica | de Reus et al. (2009) | in-situ | MDB Geophysica | 3.1.11 |
| 2006 | CR-AVE | WB-57 | Flores et al. (2006) | in-situ | NASA WB-57 | 3.1.9 |
| 2008 | CONCERT | airliner | Voigt et al. (2010) | in-situ | DLR Falcon | 3.1.12 |
| 2008 | *Fallstreaks 2008* | airliners | Atlas and Wang (2010) | lidar | Goddard | 3.2.10 |
| 2008 | *ACTA* | airliners | Vázquez-Navarro et al. (2015) | satellite | Europe & North Atlantic | 3.2.12 |
| 2011 | CONCERT2011 | airliners | Kaufmann et al. (2014) | in-situ | DLR Falcon | 3.1.12 |
| 2011 | COSIC | BAe 146 | Jones et al. (2012) | in-situ | FAAM BAe-146 | 3.1.14 |
| 2012 | *Cameras* | airliners | Schumann et al. (2013a) | cameras | Munich | 3.2.12 |
| 2014 | ML-CIRRUS | B772 or F900 | Voigt et al. (2016) | in-situ | HALO | 3.1.15 |



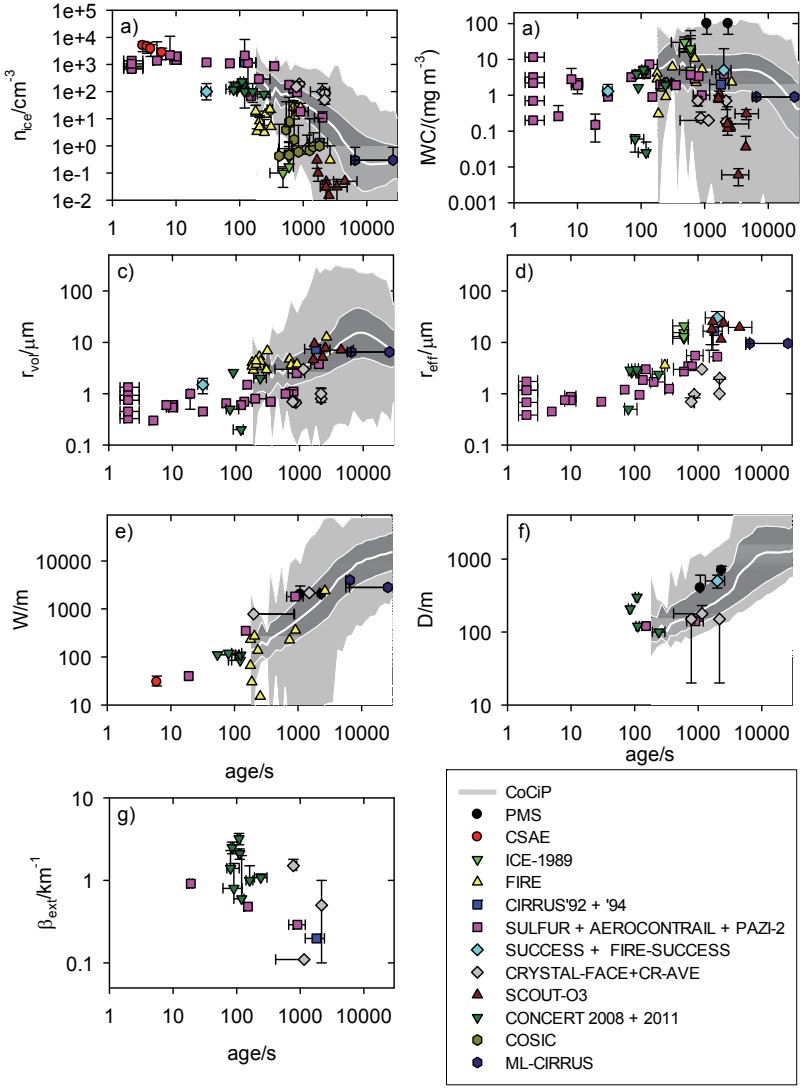

**Figure 1** Contrail parameters from in-situ observations versus contrail age, together with results from a model simulation (CoCiP). Panel a) Number concentration of ice particles $n_{ice}$ in $cm^{-3}$, b) ice water content IWC in $mg\ m^{-3}$, c) volume mean radius $r_{vol}$ in µm, d) effective radius $r_{eff}$ in µm, e) contrail width W in m, f) total contrail depth D in m, g) optical extinction $\beta_{ext}$ in $km^{-1}$. The grey-shadowed white lines depict mean CoCiP results (0, 10, 50, 90, 100 percentiles); the thick white line is the median value (50%). Various symbols denote various observation projects (or group of projects), see Table 7. Error bars, as far as available, indicate estimated ranges of variability during the measurements and reflect uncertainties (e.g., in age).





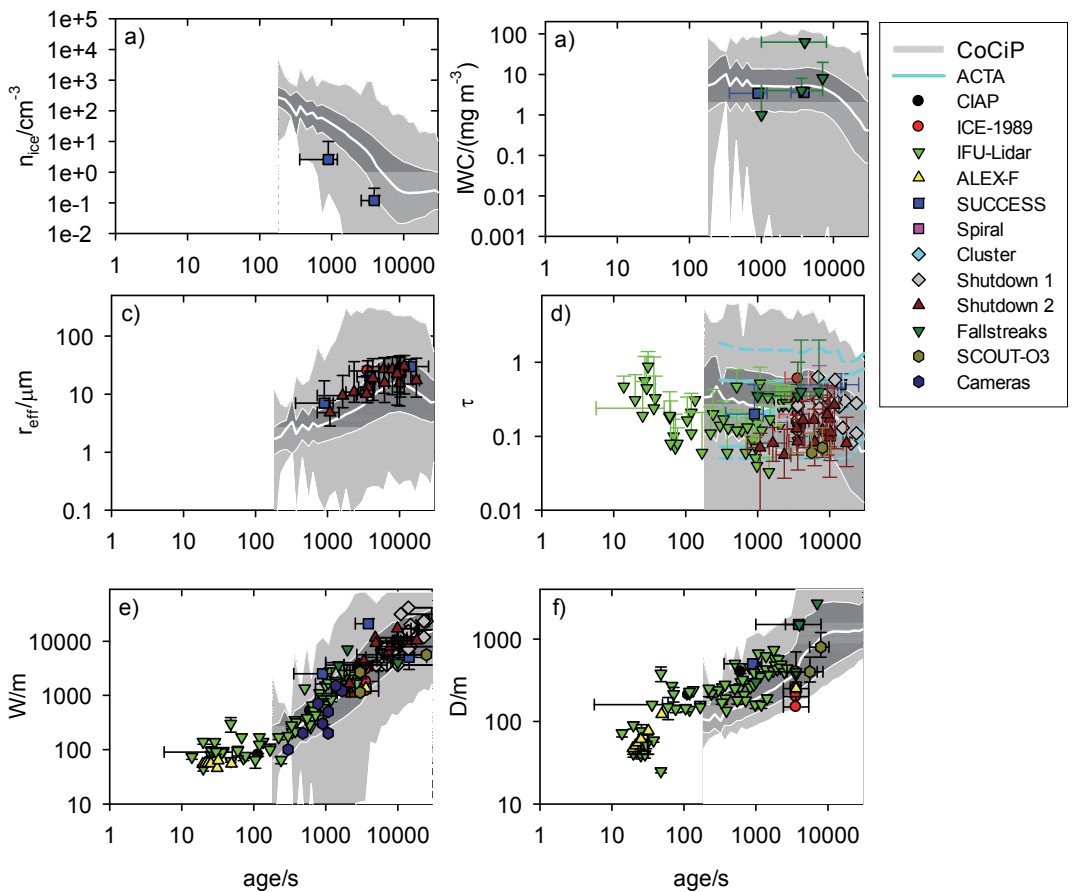

**Figure 2** Same as **Figure 1** for remote-sensing data. Panel a) ice particle concentration $n_{ice}$, b) IWC, c) effective radius $r_{eff}$, d) optical depth $\tau$ at solar wavelengths, e) width W, f) depth D. The cyan lines in panel d) depict mean values of optical depth $\tau$ derived from 25600 ACTA results (0, 10, 50, 90, 100 percentiles)




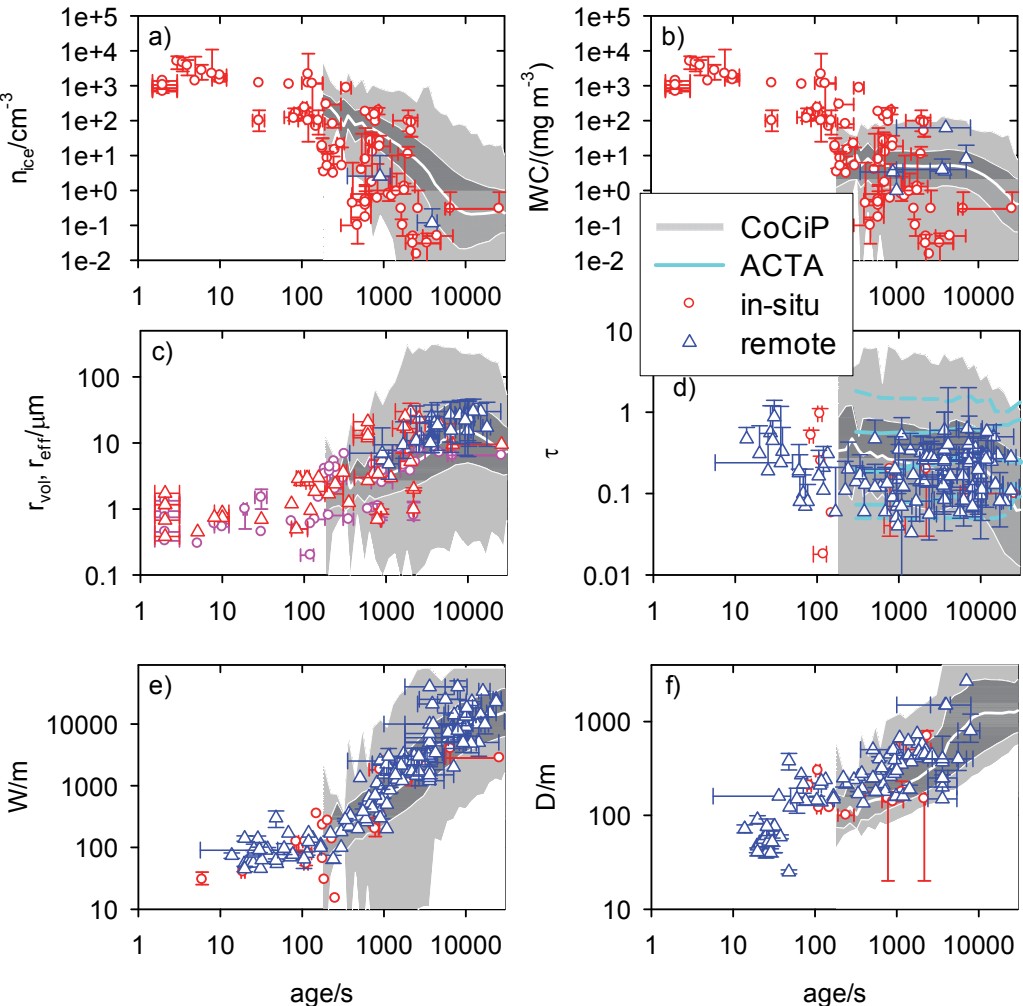

**Figure 3** Local contrail parameters from all observations collected in COLI versus contrail age, together with results from a model simulation (CoCiP percentiles) as in **Figure 1**. a) $n_{ice}$, b) IWC, c) $r_{vol}$ (purple symbols), $r_{eff}$ (red and blue symbols), d) $\tau$, e) W, f) D. The red (and purple) symbols are from in-situ measurements. The blue symbols are from remote sensing.





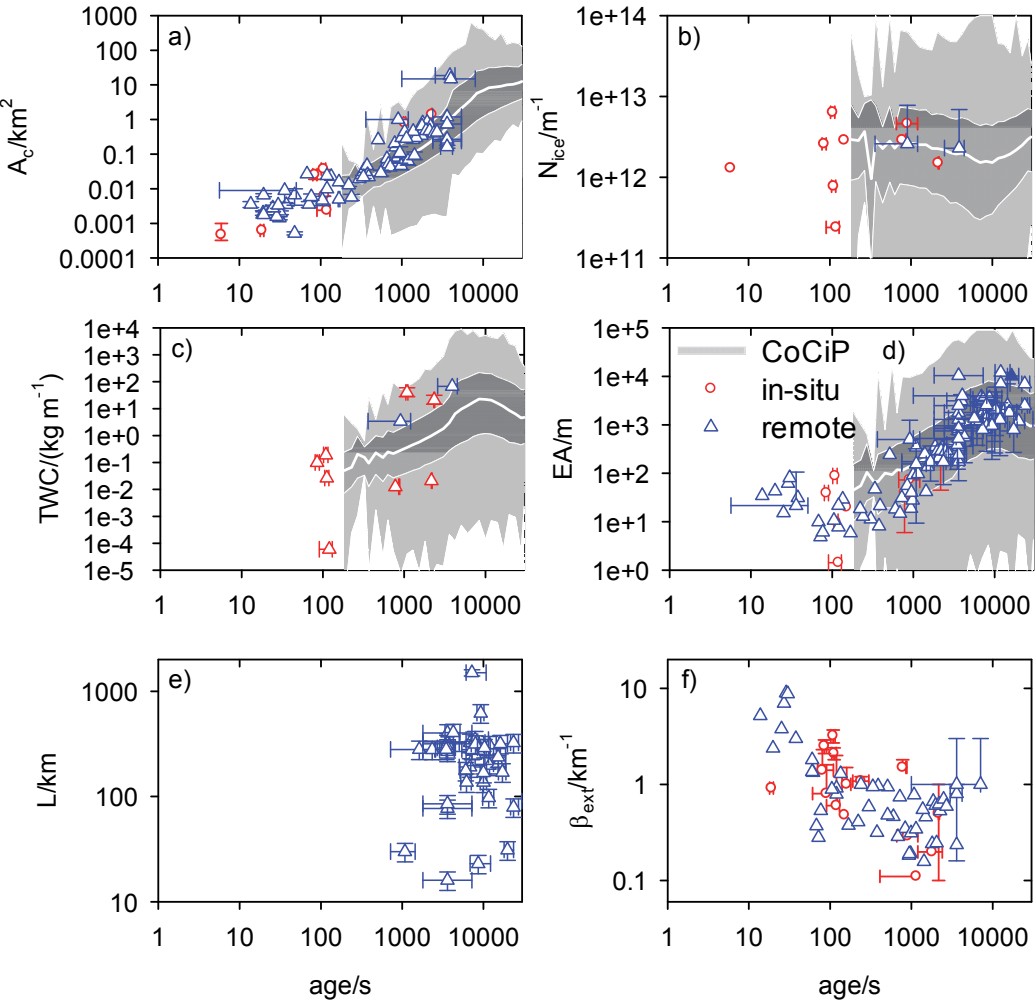

**Figure 4** Integral contrail parameters (together with length and extinction) from all observations collected in COLI versus contrail age, together with results from a model simulation (CoCiP percentiles) as in **Figure 1**. a) Contrail cross-section area $A_c$, b) total number of ice particles per length $N_{ice}$, c) total ice water content per length TWC, d) total extinction EA (= integral of optical depth over contrail width = integral of extinction over contrail area), and e) length L, f) optical extinction $\beta_{ext}$. The red symbols are from in-situ measurements. The blue symbols are from remote sensing.





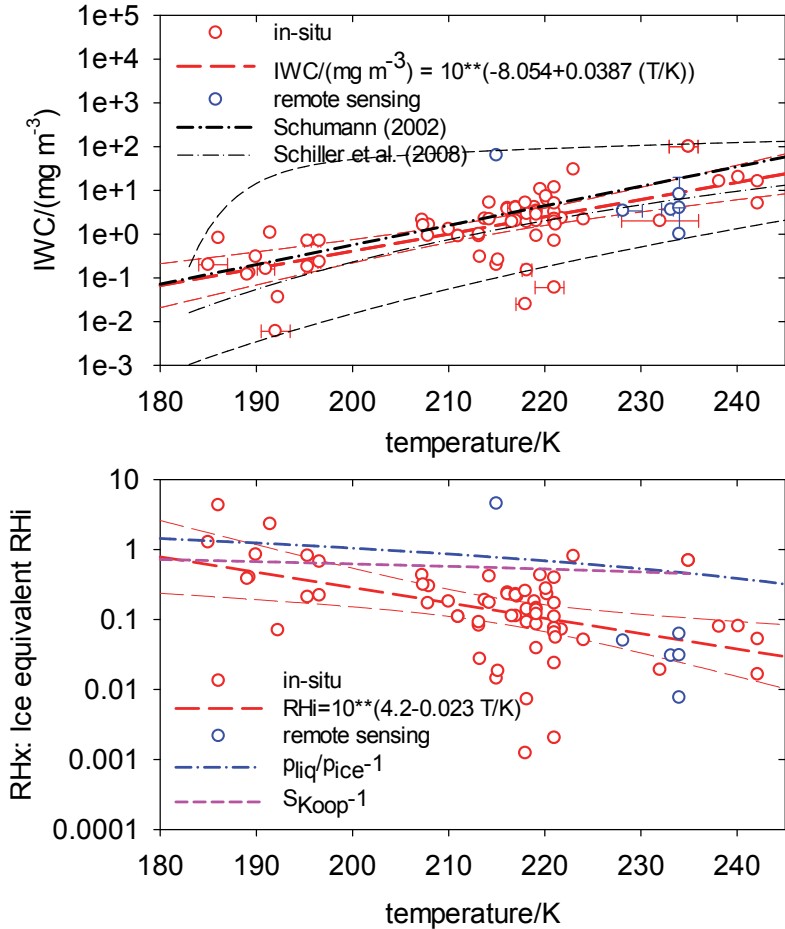

**Figure 5 Top:** Ice water content (IWC) versus temperature T from in-situ (red) and remote sensing observations (blue symbols) with a regression line for the in-situ data (red; with given formula and with 99% confidence ranges, red dashed). The thick black dash-dotted line depicts an earlier parameterization (Schumann, 2002). The thin black lines represent median (dash-dotted) and maximum/minimum values (dashed curves) from in-situ measurements in cirrus (Schiller et al., 2008). **Bottom:** Ice-equivalent relative humidity (RHx) over ice versus temperature (1 = 100%), with a regression line (red dashed, with 99% confidence ranges, thin red dashed), and with expected RHx values for ice forming from ambient supersaturation at liquid saturation (blue dash-dotted) and at the critical humidity for homogenous nucleation (Koop et al., 2000; Kärcher and Lohmann, 2002) (purple dashed line).





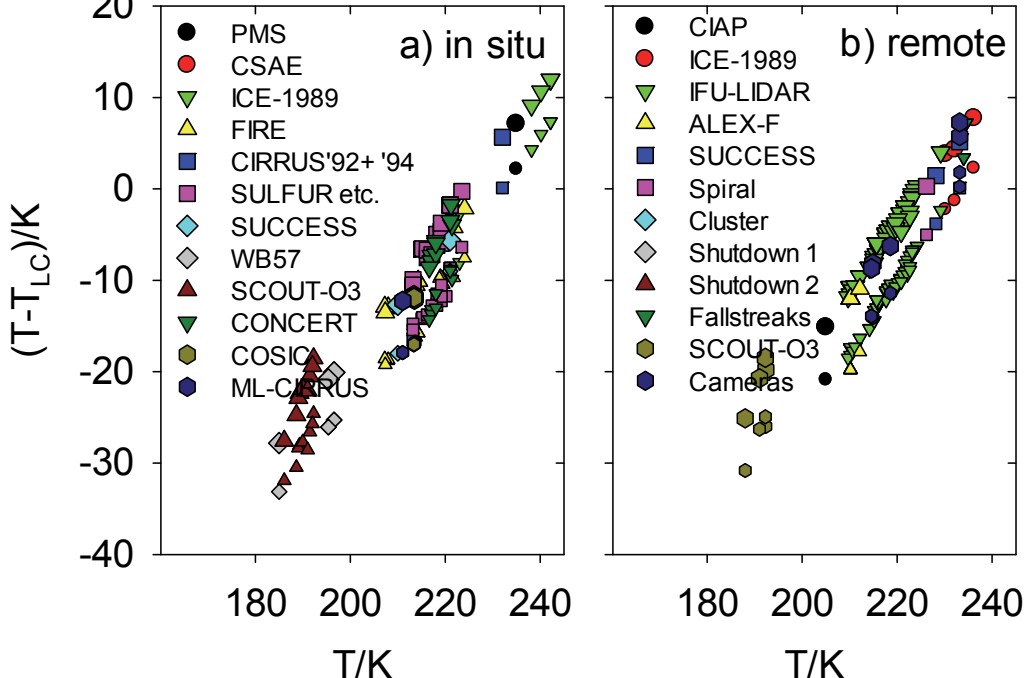

**Figure 6** Difference the between ambient temperature (T) and the threshold temperature ($T_{LC}$) versus T for a) in-situ and for b) remote-sensing contrail observations. The larger symbols depict $T$-$T_{LC}$ when $T_{LC}$ is computed for the measured or estimated humidity, the smaller symbols when computed for assumed liquid saturation. The symbols refer to the same projects as in Figures 1 and 2 (WB-57 refers to CRYSTAL-FACE and CR-AVE, which both used this aircraft) .





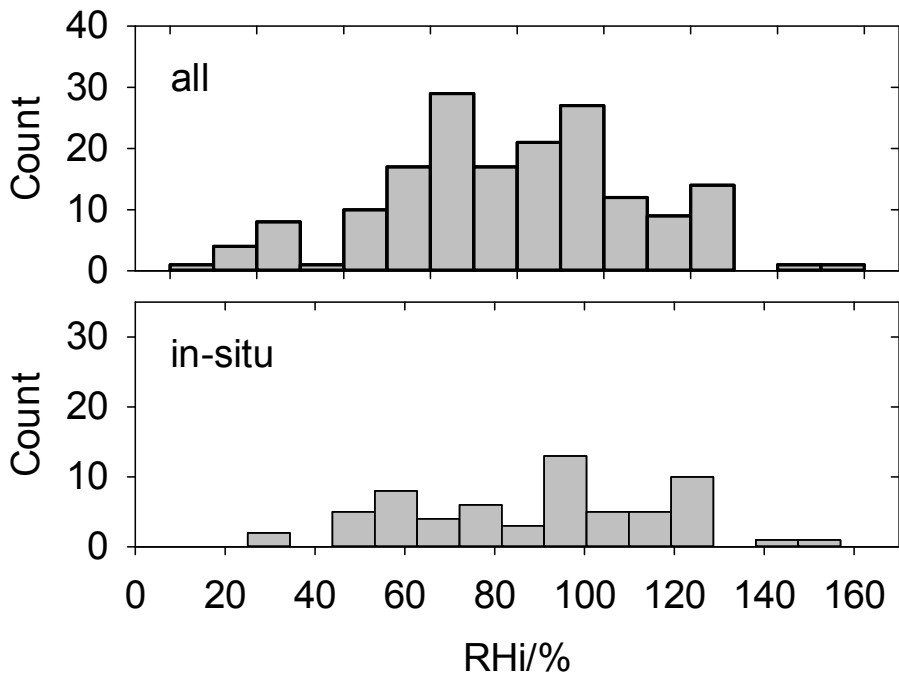

Figure 7 Histogram of relative humidity over ice (RHi) for all cases (top) and for in-situ cases (bottom).





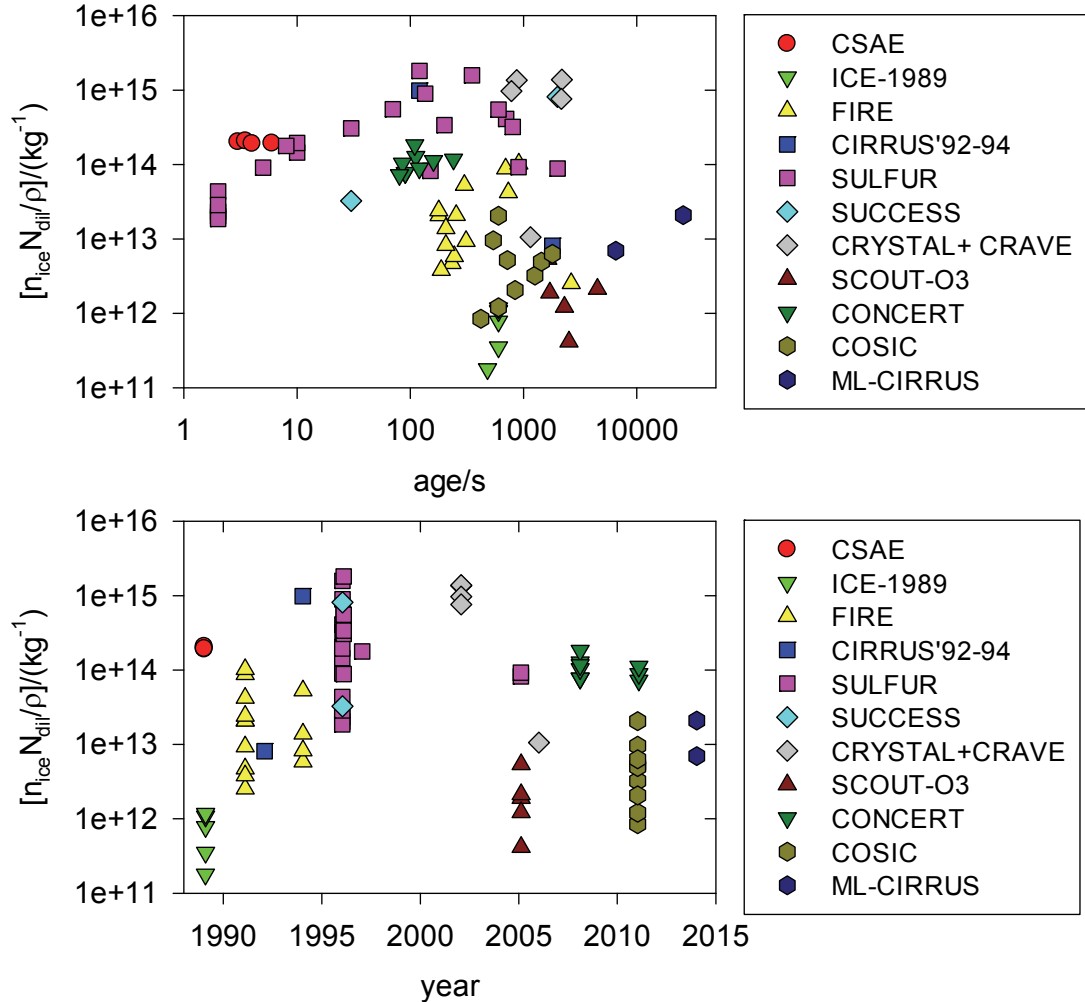

**Figure 8** Particle concentrations $n_{ice}$ normalized to density and an empirical dilution law, approximating an apparent particle emission index, $PEI_{ice}$ (the number of ice particles in the contrail per burnt fuel mass). b) Same quantity versus year of measurement.



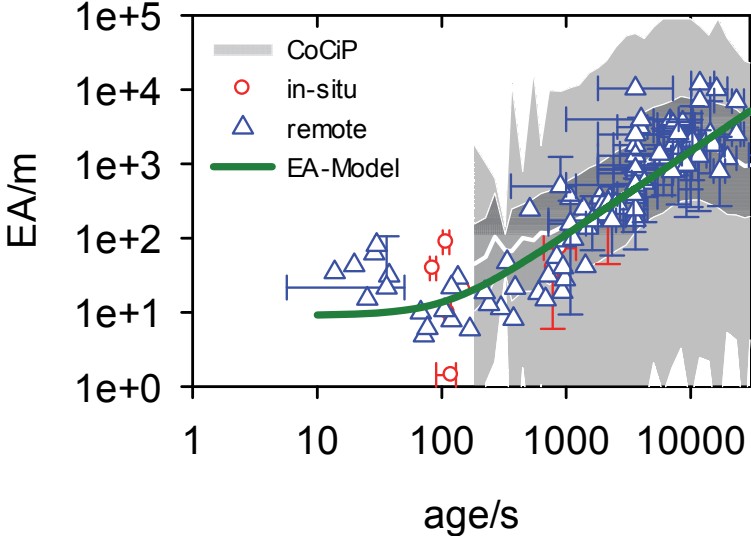

**Figure 9** Total extinction EA versus time: Data (symbols as before) and a simple model (full dark green curve).

