# Peer review of "Properties of individual contrails: A compilation of observations and some comparisons"

_Atmospheric Chemistry and Physics, 2016_

## Referee Comment (RC1) · Anonymous Referee #1 · 13 Oct 2016

This is an excellent contribution to the literature of contrail research. It could be viewed either as a review article, with a complete set of compiled data attached, or as a data library (as advertised) with detailed annotation and discussion included. In either case, there is also some additional analysis included, which extends and supplements the prior work quoted in the manuscript.

Given the extensive author list, who have presumably all extensively read this document: especially the parts where their data is discussed, there is not much that additional review can add. I only offer a few minor comments where I thought the wording was confusing, at least to me. For this very polished manuscript, these comments at least show that I read it fully and carefully, whether the authors choose to act on these comments or not.

Again, a valuable contribution, both on what it describes but also as a resource/library for future work. The compendium of older data is especially valuable. It is interesting to read the evolution of work, and how much greater detail has become accessible as our knowledge and measurement approaches have improved. Minor comments: 1) Page 9, lines 3-4 "threshold temperature is slightly higher for finite overall propulsion efficiency". I understand fully what is being meant here, but don't all engines have finite overall propulsion efficiency? Isn't the point that the change in threshold temperature is dependent on the *change* in overall propulsion efficiency? I find this sentence confusing as written, even though I believe I understand the point intended. 2) Page 10, line 9. "an undefined aircraft" ... isn't the issue that the aircraft wasn't identified? If it had been identified, then its type and characteristics would have been known (and maybe even defined, I don't know). 3) Page 16 lines 10 - 12. "The BAe146 is propelled by four turbofan engines. ... comparable jet aircraft." Most current commercial airliners use turbofan engines as well. I don't understand the point here. Is the comparison a) of the BAe146 to those few airplanes (old Learjets, some military trainers, etc.) that use pure (no bypass) jet engines? Or is the comparison b) of older technology engines on the BAe146, which might have higher PM emissions, to more modern turbofans on today's commercial fleet? The current wording suggests that the first a) comparison is to be made, but that doesn't seem very broadly interesting. The latter b) is more interesting but the wording is incorrect for this comparison. Of these minor comments, I would hope that at least comment 3) would be clarified before publication.

---

## Referee Comment (RC2) · Anonymous Referee #2 · 27 Oct 2016

This paper nicely sums up findings of several decades of contrail measurements into a comprehensive data base. The presentation of the different measurements is structured both chronologically and with respect to the applied methods (i.e. in situ or remote sensing), and thus, provides insight into the evolution of the respective measurement capabilities. The text is well written and easy to follow.

I suggest publication of the work in ACP after a few minor comments have been addressed:

- there are a number of complex formulas embedded in the text, those should be given their own line

- the paper refers to a number of grey-literature publications that I would usually

ask to be removed from the list of references but might be needed here for completeness of the data base. Maybe the authors can restrict those papers to the once that are available online or add them to the supplement? I am not entirely certain of the ACP policy in that regard.

- please check if all introduced abbreviations are useful, some of them are only used a few times while others are introduced later in the paper when the respective term had already been used. Also, please stick to abbreviations once they are introduced.

- page 3, line 8: enable instead of provide

- page 3, line 24: omit mainly

- page 11, line 20: please clarify if excluding data from the plots also means that they are not included in COLI

- page 13, line 21: within about 1 km of the tropopause?

- page 14, line 22: please revise the statement: are these possible self-encounters identified from in-situ measurements or the trajectory analysis?

- page 16, line 25: west of Shannon?

- page 19, line 4: is it correct that you combine the measurement with random flight data about 30 years later to determine contrail age? I worry about the feasibility of this approach...

- page 20, line 14: please provide details on the comparison or omit statement

- page 24, line 6: should the total extinction be introduced earlier in the paper?

- page 24, line 27: replace blowing with something less colloquial

- page 25, line 1: is there a paper that describes how those trajectories have been determined?

- page 28, line 9: CALIPSO is the satellite, CALIOP is the lidar.

- page 37, line 2: Does this mean that all CoCiP data plotted and in the table refer to calculations performed with the ACCRI data set of 2006?

- Figure 3: use the same scale for Fig 3b as for Figs 1b and 2b. It looks like the red symbols in Fig 3b are the same as in Fig 3a and not those in Fig 1b. Maybe that explains the different scale.

- Figure 6: omit the after Difference in the caption

---

## Author Comment (AC1) · 26 Nov 2016

We thank the reviewer for his positive comments on the paper and for his helpful suggestions.

We repeat the minor comments and then explain our responses:

1) Page 9, lines 3-4 "threshold temperature is slightly higher for finite overall propulsion efficiency". I understand fully what is being meant here, but don't all engines have finite overall propulsion efficiency? Isn't the point that the change in threshold temperature is dependent on the *change* in overall propulsion efficiency? I find this sentence confusing as written, even though I believe I understand the point intended.

Response: The text is changed to: The data showed that the SAC threshold temperature is slightly higher when accounting for the overall propulsion efficiency.

2) Page 10, line 9. "an undefined aircraft" ... isn't the issue that the aircraft wasn't identified? If it had been identified, then its type and characteristics would have been known (and maybe even defined, I don't know).

Response: The text is changed to: For cases A1 and U of Schröder et al. (2000) (60 and 1200 s aged contrails behind an Airbus A319 and an aircraft of unknown type),...

3) Page 16 lines 10 - 12. "The BAe146 is propelled by four turbofan engines. ... comparable jet aircraft." Most current commercial airliners use turbofan engines as well. I don't understand the point here. Is the comparison a) of the BAe146 to those few airplanes (old Learjets, some military trainers, etc.) that use pure (no bypass) jet engines? Or is the comparison b) of older technology engines on the BAe146, which might have higher PM emissions, to more modern turbofans on today's commercial fleet? The current wording suggests that the first a) comparison is to be made, but that doesn't seem very broadly interesting. The latter b) is more interesting but the wording is incorrect for this comparison.

Response: The text is changed to:

The BAe146 is propelled by four turbofan engines. It would be interesting to know whether turbofan engines emit the same number of soot particles as comparable jet engines.

---

## Author Comment (AC2) · 26 Nov 2016

We thank the reviewer for his detailed and helpful comments.

Here, we repeat his comments/suggestions for minor changes and explain our responses:

Comment: there are a number of complex formulas embedded in the text, those should be given their own line

Reply: Done

Comment: the paper refers to a number of grey-literature publications that I would usually ask to be removed from the list of references but might be needed here for completeness of the data base. Maybe the authors can restrict those papers to the

once that are available online or add them to the supplement? I am not entirely certain of the ACP policy in that regard.

Reply: According to ACP, informal or so-called "grey" literature may be referred to if there is no alternative from the formal literature. http://www.atmospheric-chemistry-and-physics.net/for_authors/manuscript_preparation.html Here, we include references to grey literature only for cases which we rate as important and where formal literature is not available. We checked with our librarians and learned that grey literature cannot be included in the supplement because of copyright restrictions. Instead, we made two of the older DLR reports (aufm Kampe 1942, and Schmidt 1941) available at http://elib.dlr.de/.

Comment: please check if all introduced abbreviations are useful, some of them are only used a few times while others are introduced later in the paper when the respective term had already been used. Also, please stick to abbreviations once they are introduced.

Reply: Done at several places.

Comment: page 3, line 8: enable instead of provide

Reply: done

Comment: page 3, line 24: omit mainly

Reply: reworded

Comment: page 11, line 20: please clarify if excluding data from the plots also means that they are not included in COLI

Reply: The data are not included in COLI. The sentence is now deleted to avoid misunderstanding.

Comment: page 13, line 21: within about 1 km of the tropopause?

Reply: Deleted

Comment: page 14, line 22: please revise the statement: are these possible self-encounters identified from in-situ measurements or the trajectory analysis?

Reply: Text clarified; "From" replaced by "for"

Comment: page 16, line 25: west of Shannon?

Reply: text reordered to avoid misunderstanding

Comment: page 19, line 4: is it correct that you combine the measurement with random flight data about 30 years later to determine contrail age? I worry about the feasibility of this approach...

Reply: The reviewer is right; that part does not help to improve the estimate and actually is not used. Text deleted.

Comment: page 20, line 14: please provide details on the comparison or omit statement Reply: we now say that the data fit into the range of other results.

Comment: page 24, line 6: should the total extinction be introduced earlier in the paper? Reply: done

Comment: page 24, line 27: replace blowing with something less colloquial

Reply: Done (deleted)

Comment: page 25, line 1: is there a paper that describes how those trajectories have been determined?

Reply: The method is standard (e.g., used in CoCiP) and has been described in the text. Therefore, no further reference is needed.

Comment: page 28, line 9: CALIPSO is the satellite, CALIOP is the lidar. Reply: rephrased to: the lidar CALIOP on CALIPSO
**Interactive comment**

Comment: page 37, line 2: Does this mean that all CoCiP data plotted and in the table refer to calculations performed with the ACCRI data set of 2006?

Reply: Yes. This part is rewritten to avoid misunderstanding.

Comment: Figure 3: use the same scale for Fig 3b as for Figs 1b and 2b. It looks like the red symbols in Fig 3b are the same as in Fig 3a and not those in Fig 1b. Maybe that explains the different scale.

Reply: Thank you for noting the plot error; now corrected.

Comment: Figure 6: omit "the" after Difference in the caption

Reply: Done

――――――――――――――――――――

---

## Author Response (AR1)

**To the Editor:**

We thank both reviewers for their positive and helpful comments. We have taken all suggestions for minor revisions into account, as explained in the two responses to the two reviewers.

In the last 2 weeks, we performed further checks which resulted into a few minor changes in the paper, as can be seen from the text changes identified by MS-Word on the coming pages. The figures have been adapted accordingly. No major change.

We note that the paper on the Geophysica contrail did appear as ACPD paper (Schumann et al., 2016).

Also, the paper which we cited from the AMS Meteorological Monographs (Schumann and Heymsfield, 1916) was accepted some time ago and is now available online as well.

We think that our paper is ready for publication and look forward to your decision.
It would be nice if the paper could appear in 2016.

Best regards,

Ulrich Schumann
on behalf of all co-authors.
26 Nov 2016.

[revised manuscript text omitted]

---

## Author Response (AR2)

Dear editor,

I am pleased to hear that our paper got accepted for publication in Atmospheric Chemistry and Physics.

I have changed the sequence of references as requested.

There are no other changes.

I thank again the editors and the reviewers for their work.

With kind regards,

Ulrich Schumann

Telephone +49 173 28 75 353

18 December 2016.